# Wind waves in the North Atlantic from ship navigational radar: SeaVision development and its validation with Spotter wave buoy and WaveWatch III

Natalia Tilinina[1,2], Dmitry Ivonin[1], Alexander Gavrikov[1], Vitali Sharmar[1], Sergey Gulev[1,3], Alexander Suslov[1], Vladimir Fadeev[4], Boris Trofimov[4], Sergey Bargman[4], Leysan Salavatova[5], Vasilisa Koshkina[5], Polina Shishkova[1], Elizaveta Ezhova[5], Mikhail Krinitsky[1], Olga Razorenova[1], Klaus Peter Koltermann[6], Vladimir Tereschenkov[1] and Alexey Sokov[1]

[1] Shirshov Institute of Oceanology, RAS, Nakhimovsky ave. 36, 117997, Moscow, Russia

[2] Université Grenoble Alpes, CNRS, IRD, Grenoble-INP, Institut des Géosciences de l'Environnement, 70 rue de la Physique, 38400, Grenoble, France

[3] A.M. Obukhov Institute of Atmospheric Physics, RAS, Pyzhevskiy Lane 3, Moscow, 109017, Russia

[4] Joint stock company "Marine Complexes and Systems", Aleksandrovskoy Fermy ave. 2 office 2H, 192174, Saint-Petersburg, Russia

[5] Moscow Institute of Physics and Technology, Institutskiy Pereulok 9, 141701, Dolgoprudny, Moscow Region, Russia

[6] Lomonosov Moscow State University, Moscow Russia

*Correspondence to*: Natalia Tilinina (tilinina@sail.msk.ru) and Alexander Gavrikov (gavr@sail.msk.ru)

**Abstract**

Wind waves play an important role in the climate system, modulating the energy exchange between the ocean and the atmosphere and effecting ocean mixing. However, existing ship-based observational networks of wind waves are still sparse limiting therefore the possibilities of validating satellite missions and model simulations. In this paper we present data collected on three research cruises in the North Atlantic and Arctic in 2020 and 2021 and the SeaVision system for measuring wind wave characteristics over the open ocean with a standard marine navigation X-band radar. Simultaneously with the SeaVision wind wave characteristics measurements we also collected data from the Spotter wave buoy at the same locations and we ran the WaveWatch III model in a very high-resolution configuration over the observational domain. SeaVision measurements were validated against co-located Spotter wave buoy data and intercompared with the output of WaveWatch III simulations. Observations of the wind waves with the navigation X-band radar were found to be in good agreement with buoy data and model simulations with the best match for the wave propagation directions. Supporting datasets consist of significant wave heights, wave directions, wave periods and wave energy frequency spectra derived from

both SeaVision and the Spotter buoy. All supporting data are available through the PANGAEA repository – https://doi.pangaea.de/10.1594/PANGAEA.939620 (Gavrikov et al., 2021). The dataset can be further used for validation of satellite missions and regional wave model experiments. Our study shows the potential of ship navigation X-band radars
(when assembled with SeaVision or similar systems) for the development of a new near-global observational network providing a much larger amount of wind wave observations compared to e.g. Voluntary Observing Ship (VOS) data and research vessel campaigns.

## 1 Introduction

Ocean wind waves play a critically important role in air-sea energy and gas exchanges (Gulev and Hasse 1998; Andreas et
al. 2011; Blomquist et al. 2017; Ribas-Ribas et al. 2018; Cronin et al. 2019; Xu et al. 2021 among many others) and in ocean surface mixing (McWilliams and Fox-Kemper 2013; Buckingham et al. 2019; Studholme et al. 2021), thus being an important active component of the coupled climate system (Cavaleri et al. 2012; Fan and Griffies 2014). At the same time, massive long-term observations of wind waves over global oceans still have insufficient coverage and quality compared to other surface variables (e.g. air and sea surface temperatures). Wind waves are wind-driven ocean surface gravity waves.
Visual wave observations from Voluntary Observing Ships (VOS) while providing the longest time coverage (formally going back to the mid- 19th century) suffer from space- and time- dependent sampling biases as well as from both random and systematic biases, and require continuous validation (Gulev et al. 2003). Remote sensing datasets of wind waves go back to 1985 (Ribal and Young, 2019), when the first satellite radar altimeter missions (Seasat in 1978 (the first satellite to provide data) and Geosat in 1985) were launched and started to provide ocean surface elevations with high temporal and
spatial resolution. However, remote sensing data have to be validated against *in situ* measurements, typically available from buoys (such as NDBC buoys, Swail et al. 2010 or NOWPHAS, Nagai et al. 2005). Buoys measure vertical and horizontal displacements of the ocean surface (such as Spotter or Datawell buoys with up to 2.5 Hz sampling frequency, Raghukumar et al., 2019) and provide highly accurate estimates of wind waves characteristics, effectively now assimilated into Numerical Weather Prediction (NWP) models. Assimilation of the significant wave heights form wave buoys in operational wave
model decreases root-mean-square error in significant wave height forecasts by 27% on average (Smit et al., 2021). However, buoy networks are sparse with most deployments being in the coastal regions and can only effectively serve for verification of all other datasets rather than for developing global or regional climatologies.

Starting from the 1980s, considerable progress in wind wave modelling (WAMDI, 1988; Hasselmann et al., 1985; Cavaleri et al., 2020) resulted over the last decade in the development of multiple global and regional wind wave hindcasts generated
by spectral wave models such as WAM (WAMDI 1988) or WAVEWATCH (WW3DG 2019) forced by atmospheric reanalyses or climate models, and providing multidecadal wind wave fields with high temporal and spatial resolution (Casas-Prat et al., 2018; Semedo et al., 2018; Morim et al. 2020, 2022; Sharmar et al. 2021 among others). Being currently a widely accepted source for estimating long-term climate variability in wave characteristics, wind wave hindcasts also suffer from

the inaccuracy in the modelling of many aspects of wind wave dynamics, including e.g. extremely high wave peaks at high wind speeds during the storm passage (Cavaleri et al., 2020).

Summarizing, all three sources of global wind wave information (VOS, satellite data and model hindcasts) require data for extensive validation. Existing wave buoys deployed in a few locations cannot solve this problem to the full extent. Thus, investigating alternative sources of massive wind wave data remains a challenge. In this respect, ship navigation radars represent an option whose potential, especially in open ocean regions, is not yet explored to its fullest extent. Here, we present the results of the development and validation of the SeaVision system for wind wave observations in the open ocean using standard navigation marine X-band radars which allows for real time monitoring of wind wave characteristics along the commercial ship tracks.

Applicability of the navigation radars for measurements of the wind wave characteristics was first noted by Young et al. (1985). Radar images of the ocean surface, known as sea clutter, are generated by the Bragg scattering (Crombie, 1955) of the electromagnetic signal by the ripples on the ocean surface produced by the wind. Being emitted from the radar, an electromagnetic signal reaches the ocean surface and further, being reflected by ripples on the ocean surface, and is received back by the radar antenna when the ocean surface is rough enough (i.e. ripples are developed). Under a wind speed of > 3 m/s and waves height > 0.5 m, the surface waves field becomes detectable on the radar image of the sea clutter (Hatten et al., 1998; Hessner and Hanson, 2010). Time sequences of these images are further analyzed for estimating the wind wave characteristics. The associated retrieval procedures can be based on various approaches, which include signal-to-noise ratio derived from the image spectrum (Nieto-Borge et al., 1999; Neito-Borge et al., 2008; Seemann et al., 1997), statistical analysis of the island-to-trough ratio on the sea clutter images (Buckley and Alter, 1997; Buckley and Alter, 1998), analysis of the image texture (Gangeskar, 2000), wavelet technique (Huang and Gill, 2015), the least square approach (Huang et al., 2014) and shadowing analysis (Gangeskar, 2014). Methodologies may also be based on the combination of these methods with the use of artificial neural networks (Vicen-Bueno et al., 2012). This may also include the analysis of the Doppler shift of the received radar signal that is based on the well-defined relationship between orbital velocities and wave height for linear gravity waves (Plant, 1997; Plant et al., 1987; Johnson et al., 2009; Karaev et al., 2008; Hwang et al., 2010; Hackett et al., 2011; Chen et al., 2019). There are many aspects of the sea clutter radar images analysis: Nieto Borge and Guedes Soares (2000) for example proposed an approach considering superpositions of swell and wind sea components, that allowed to derive wind waves and swell contributions to the total wave field, along with directional characteristics. There are also attempts to use images of the sea clutter revealed from X-band radars for estimating the current-depth profiles with a Eulerian approach (Campana et al., 2017), to retrieve wind speed and wind direction (Chen et al., 2015; Dankert and Horstmann, 2007; Dankert et al., 2003; Vicen-Bueno et al., 2013), and to derive surface characteristics (Senet et al., 2001).

On this basis, several commercial systems such as WaMoS II (http://www.oceanwaves.de), SeaDarQ (Greenwood et al., 2018) and WaveFinder (Park et al., 2006) were developed. The most widely used system nowadays is WaMoS II (software and hardware details provided in Reichert at al., 1999); it is focused on the operational monitoring of the sea state (wind

waves and surface currents) and operational management of oil platforms and ships using nautical X-band radars. Derkani et al. (2021) provided a dataset of the wind, waves and surface currents over the Southern Ocean collected with WaMoS II. In combination with other sources of the data (altimetric wave radar, vessel hydrodynamic simulator), wind waves estimates

from navigational radar can be used to manage security of the offshore systems, assess ship fatigue due to mechanical environmental influence (Drouet et al., 2013), or for the real-time prediction of ship rolling (Hilmer and Thornhill, 2015).

We present the design and pre-processing methodology of the SeaVision system along with the dataset collected during three research cruises (Figure 1). SeaVision was developed in collaboration between the Shirshov Institute of Oceanology of the Russian Academy of Sciences (IORAS, https://ocean.ru/) and the Joint stock company "Marine Complexes and Systems"

("MC&S" J.S.C., https://www.mcs.ru/). SeaVision is developed on the basis of the sea ice monitoring system with navigational marine radar – IceVision (https://ice.vision/en). The pilot version of SeaVision was tested and validated in two North Atlantic cruises in 2020 and 2021 and in the Arctic cruise in 2021 (Figure 1). The major advantage of the presented dataset is the provision of the co-located Spotter wave buoy data with SeaVision records at almost 50 locations and outputs of WaveWatch III (WW3) model experiments forced by ERA5 reanalysis (Hersbach et al., 2020) for the corresponding

domains. We present in this study the SeaVision system and dataset of the measurements of the wind waves in the open ocean and its comprehensive analysis.

The paper is organized as follows. In Section 2 we provide details of the research cruises, technical specifications of the SeaVision system, data collection and analysis principles as well as the description of the WW3 model setup. Section 3 presents the results of the analysis and validation of the SeaVision dataset against Spotter buoy data and the comparison with

the WW3 model output. The concluding section 4 summarizes the results and discusses the perspectives of the use of ship navigation radars for a massively enhanced collection of wind wave information in the open ocean.

## 2 Data collection and analysis

We provide definitions of all parameters included in the published dataset in Appendix C. For wind and wave directions we use meteorological convention implying that both wind and waves are coming from the specified direction (blow into

compass).

### 2.1 Ship cruises

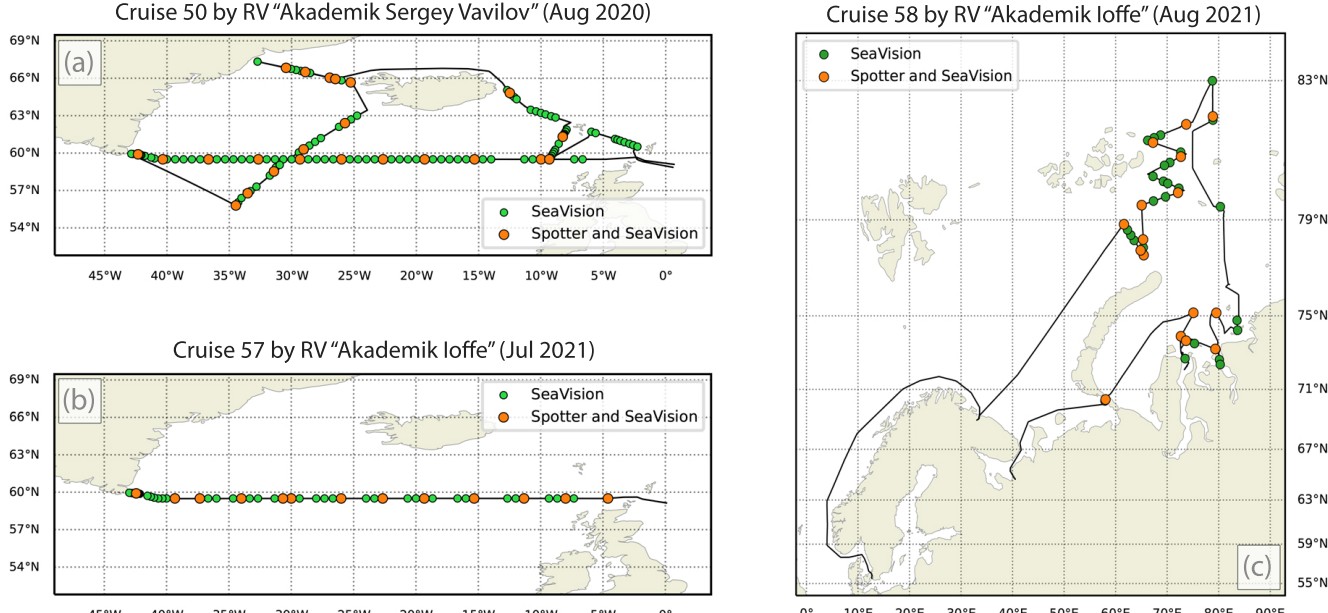

**Figure 1: Ship tracks of the three cruises of the research vessels (R/V) "*Akademik Sergey Vavilov*" (a) and "*Akademik Ioffe*" (b,c). Green dots indicate locations where only SeaVision radar data were collected, orange dots show the locations for which SeaVision records were co-located with Spotter wave buoy measurements. Cruise numbers are counted from the beginning of the R/V operation.**

Figure 1 demonstrates ship tracks of the three research cruises, during which wind wave data were collected. Research cruises were carried out by IORAS research vessels (R/Vs) *"Academik Sergey Vavilov"* and *"Academik Ioffe"*. Table 1 provides a general information about the cruises and detailed information on the coordinates and dates and is provided in Appendix A. The two cruises in the subpolar North Atlantic (Figure 1a, b) were focused on the regular survey of the 59.5°N oceanographic trans-Atlantic cross-section and cross-sections in the Denmark Strait (Verezemskaya et al., 2021). During these cruises the R/V makes full-depth CTD profiling. The distances between the hydrographic stations vary from ~30 km in the open ocean to a few kilometers near the East Greenland coast with the time allocated for each station (ship is drifting) varying from 2 to 6 hours. Here and later in the manuscript we determine stations as the locations where wind wave observations were carried out (Table A1). Between the stations the R/V travels at a speed of approximately 6 to 10 kn. During the cruise of R/V *"Academik Ioffe"* in the Kara Sea (Figure 1c), stations were somewhat shorter in time (2-3 hours). During all cruises wave observations were carried out after completing hydrographic profiling. For operating solely SeaVision, the R/V position was strictly stationary being controlled by bow and stern thrusters of the R/V. When SeaVision was used together with the free drifting Spotter buoy, the thrusters were off to provide also free drifting of the R/V. This allowed for measurements of the background wave field by both SeaVision and the Spotter buoy. At each station we first

released the Spotter buoy with a supplementary floating buoy dumping cable vibrations. Such design allows for the maintenance of at least 300 m distance between the buoys and the R/V. Then, both buoys were in the free-floating mode for at least 30 min during which the recording was performed by both SeaVision and the Spotter buoy (Figure 4a). Lastly, both buoys were pulled back onboard. The Spotter buoy measured vertical and horizontal displacements starting from its release until being retrieved back onboard. After completing measurements at each station, only the data recorded during the free-floating mode were used for the joint analysis of SeaVision and Spotter buoy records. During all SeaVision and Spotter buoy measurements, standard meteorological parameters were measured using the onboard meteostation.

**Table 1: Research cruises during which the wind waves observations were carried out by research vessels (R/Vs) "*Akademik Sergey Vavilov*" (ASV) and "*Academik Ioffe*" (AI). Adjacent numbers in the first column correspond to the R/V cruise numbers counted from the beginning of the R/V operation.**

| Cruise | Start date and location | End date and location | Distance sailed | Number of stations (with Spotter buoy) |
|---|---|---|---|---|
| ASV50 | 08/08/2020 Kaliningrad Russia | 08/09/2020 Kaliningrad Russia | 10465 km | 21 |
| AI57 | 27/06/2021 Kaliningrad Russia | 02/08/2021 Kaliningrad Russia | 7745 km | 11 |
| AI58 | 08/08/2021 Arkhangelsk Russia | 06/09/2021 Kaliningrad Russia | 10611 km | 16 |

**2.2 SeaVision system**

**2.2.1 Ship navigation radar signal retrieval and preprocessing**

Development of the SeaVision system was based on a commonly accepted approach of the recording and analysis of the sea clutter images. Using a similar approach, commercial systems such as WaMoS II (http://www.oceanwaves.de), SeaDarQ (Greenwood et al., 2018) and WaveFinder (Park et al., 2006) were developed. These commercial systems provide customers with their original software and hardware. In our approach we are focused on the development of an independently operating, low-cost and easy to install system compatible with the existing ship navigation radars.

Research vessels "*Akademik Sergey Vavilov*" (R/V *ASV*) and "*Akademik Ioffe*" (R/V *AI*) are equipped with the standard navigation X-band radars JRC JMA-9110-6XA and JMA-9122-6XA. Technical details of radar transmission and backscattering characteristics are given in Table 2. Both radars operate at 9.41 GHz frequency (wavelength ~3 cm), and are equipped with a 6 feet antenna with the directional horizontal resolution of 1.2° (Table 2). Radars can optionally operate at the pulse lengths of 0.08 µs, 0.25 µs, 0.5 µs, 0.8 µs, 1.0 µs. For our purposes we used the smallest possible pulse length of

0.08 μs (at the so-called "short-pulse" mode - SP1), providing the highest possible resolution of the image (thus the best resolution of the ocean surface). Our X-band radars are characterized by a 3.18 cm wavelength of the emitted electromagnetic waves (Table 2). The pulse length is the emission time of the wave beam, thus, the number of the emitted waves and the area of reflection at the ocean surface (defining spatial resolution) increase with increasing pulse length.

The SeaVision system (Figure 2) is connected to the radar via a splitter. It provides digitizations and further recording of the directionally stabilized (northward) radar sea clutter image resulting from each single full turn of the radar antenna. By doing this, SeaVision converts the sea clutter image into a digital format and records the data onto the external storage. SeaVision is also connected to the ship navigation package and simultaneously records geographical coordinates from GPS, speed over ground (SOG), and course over ground (COG). Each full turn of antenna results in ASCII file (~16 MB) consisting of a 4096x4096 matrix (1.875 m discretization at 4096 beam directions) representing the sea clutter digitized image with GPS information, SOG and COG in the file header. These files are further consolidated and converted into NetCDF format at the post-processing stage.

**Table 2: JRC JMA-9110-6XA radar (*R/V ASV*) and JMA-9122-6XA (*R/V AI*) transmission and reception characteristics.**

| Research vessel | *"Akademik Sergey Vavilov"* | *"Akademik Ioffe"* |
|---|---|---|
| Radar type | JRC JMA-9110-6XA | JMA-9122-6XA |
| Radar frequency/Wave length | 9.41 GHz / 3.18 cm | 9.41 GHz / 3.18 cm |
| Antenna rotation speed | 27 rpm | 24 rpm |
| Impulse power | 10 kW | 25 kW |
| Antenna size | 6 ft | 6 ft |
| Pulse length mode | 0.08 μs (short pulse) | 0.07 μs (short pulse) |
| Analog-digital converter (ADC) frequency/size of output matrix for one antenna turn | 80 MHz / 4096x4096 | 80 MHz / 4096x4096 |
| Azimuthal coverage/resolution | 0 – 360°/1.2° | 0 – 360°/1.2° |

| | | |
|---|---|---|
| Distance range | 231.5 – 2778 m | 231.5 – 2778 m |
| Range resolution | 12 m | 10.5 m |
| Analog-digital converter (ADC) frequency/size of output matrix for one antenna turn | 80 MHz / 4096x4096 | 80 MHz / 4096x4096 |
| Calibration coefficients A and B | A = -0.4042, B = 1.0034 | A = -0.4042, B = 1.0034 |

185

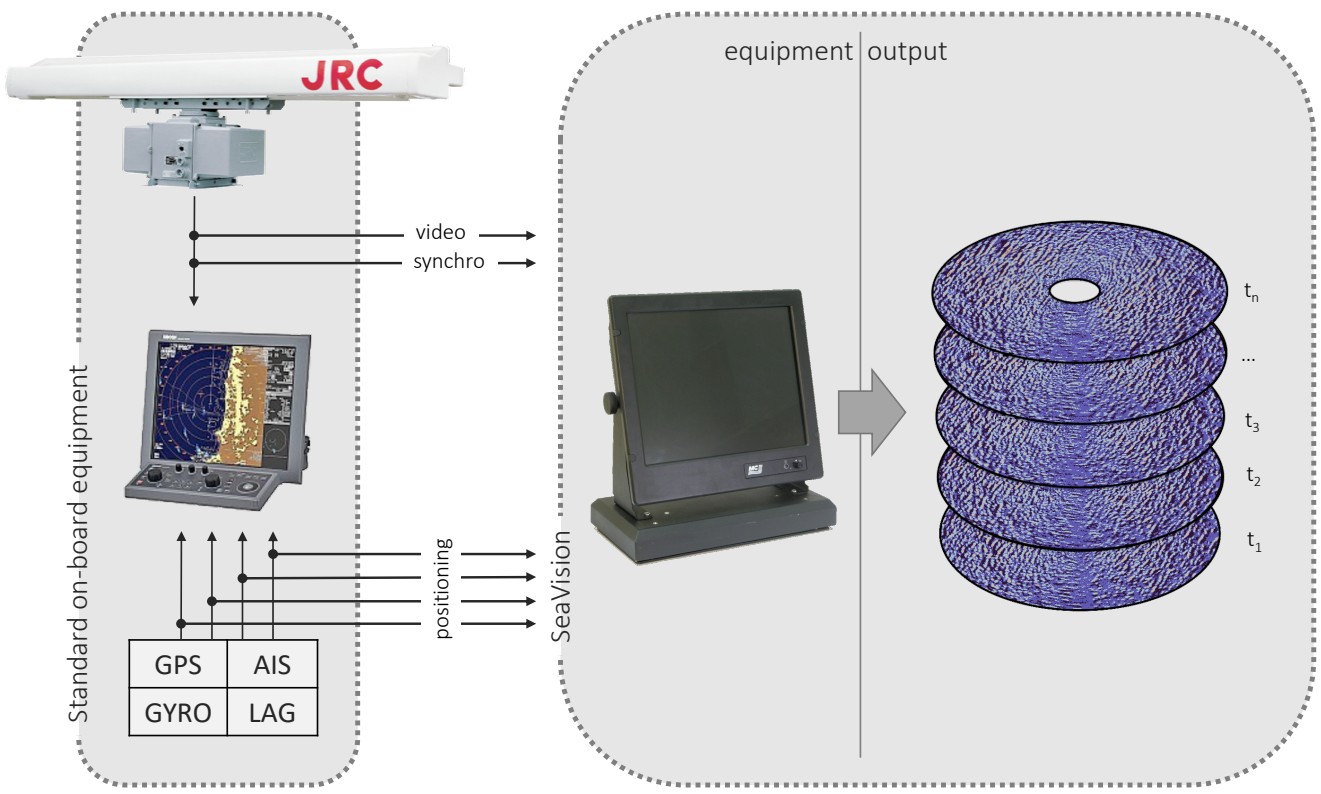

**Figure 2: SeaVision integration to the ship's navigational equipment together with an example of the series of the geographically stabilized (northward) sea clutter images, one for each antenna turn (right column). Image of the JRC radar scanner (top left) is taken from the http://www.jrc.co.jp/eng/index.html.**

190   **2.2.2 Analysis of the sea clutter images**

After the sea clutter images are collected and digitized, the next step is the postprocessing focused on the computation of significant wave height ($H_s$), wave period ($T_{m01}$), wave energy spectrum ($S_w$) and wave direction ($D_s$). Here we provide a short condensed description of the algorithm with the full details given in Appendix B. The subset collected at each station (Figure 1) consists of the 20-minute SeaVision record, which is equivalent to at least 540 images of the sea clutter (27 antenna full turns per minute for JRC JMA-9110-6XA radar).

The methodology for estimation of wind wave characteristics relies on a well established Fourier Transform (FT) technique (Nieto-Borge and Guedes Soares, 2000; Borge et al., 2004; Borge et al., 2008 among others). For each station, preprocessing of the data begins with the choice of the processing squared area (squared area of 720×720 m). For now, we locate the processing area visually by taking the area of the most apparent wave signal in the image and requiring this area to be distanced from the ship by 300 m to avoid a potential impact of the ship on the wave field and the effects of the reflection and modulation of the radar signal by the ship superstructure. When the processing area is selected, we consolidate the data captured in this area from all 540 images for further analysis. Note that the data initially sampled in polar coordinates are re-gridded at this step to a Cartesian grid of 384x384 grid points with 1.875 m spatial resolution for each subset.

The sequence of 540 matrices with 384×384 grid points each is then split into 16 sectors (22.5° width each). Further, to obtain the directional spectra estimates, we transformed the data into a 3D spectral domain by using the Fourier transform and applying the Welsh method with a half-width overlapping Hanning window (48 points, Figure 3). This returns for each sector, the three-dimensional spectrum $S_{3d,image}(k_x, k_y, f)$, where $f = \omega/2\pi$ is the frequency (Hz) and $\omega$ is the angular frequency, $k_x$ and $k_y$ (rad/m) are the components of the wave vector $\vec{k}(k_x, k_y)$. Then, for each sector we capture the spectrum power within the band along the line satisfying the linear dispersion relation for ocean waves (Figure 3):

$$\omega = \sqrt{gk} + kU cos\theta , \tag{1}$$

where $k$ is the wave number (rad/m), $g$ is gravity (m/s²), $U$ is the surface velocity (m/s) which includes surface current velocity and ship drift, and $\theta$ is the angle between the wave vector $\vec{k}$ and velocity vector $\vec{U}$. This procedure is applied to the bands corresponding to the first and the second spectral harmonics (see Appendix B for the definition of band width). The spectral power outside the bands for the two harmonics is assumed to be a background speckle-noise ($\omega_{speckle}$) (Kanevsky, 2009). Integrated spectral power outside of the bands matching the wave dispersion relation (1) is further used for estimating of the signal-to-noise ratio (SNR) as described in Appendix B and outlined in many works (Nieto-Borge et al., 1999; Hessner et al., 2002; Young et al 1985, Nieto-Borge and Guedes Soares 2000, Ivonin et al. 2016). Following Nieto-Borge et al. (1999, 2004) SNR is then converted to significant wave height $H_{s,SeaVision}$ using the linear regression equation:

$$H_{s,SeaVision} = A + B \sqrt{SNR}, \tag{2}$$

where $A$ and $B$ are empirical calibration coefficients which are specific for each radar. In this study, these coefficients were computed by fitting a linear regression (2) to the significant wave height measured by the Spotter wave buoy. Derived numerical values of A and B coefficients are given in Table 2 for both X-band radars. Wave period $T_{m01,SeaVision}$ was estimated conventionally using zeroth and first spectral moments:

$$T_{m01,SeaVision} = \frac{m_{0,SeaVision}}{m_{1,SeaVision}}, \tag{3}$$

$$m_{0,SeaVision} = \int_0^\infty S_{w,SeaVision}(f)df, \tag{4}$$

$$m_{1,SeaVision} = \int_0^\infty S_{w,SeaVision}(f)fdf \tag{5}$$

where $S_{w,SeaVision}(f)$ is the estimate of the wave energy spectrum from SeaVision:

$$S_{w,SeaVision}(f) = \left(\frac{H_{s,SeaVision}}{H_{s,image}}\right)^2 S_{image}(f) \tag{6}$$

and $H_{s,image}$ is $H_{s,image} = 4\sqrt{m_{0,image}}$ , thus being the estimate of significant wave height using the raw sea clutter image before calibration.

We note that local weather conditions, specifically rain events, can potentially affect the electromagnetic radar signal as the raindrops absorb and scatter radar signal. However, the analysis of current weather has shown that no rain events were observed during observations.

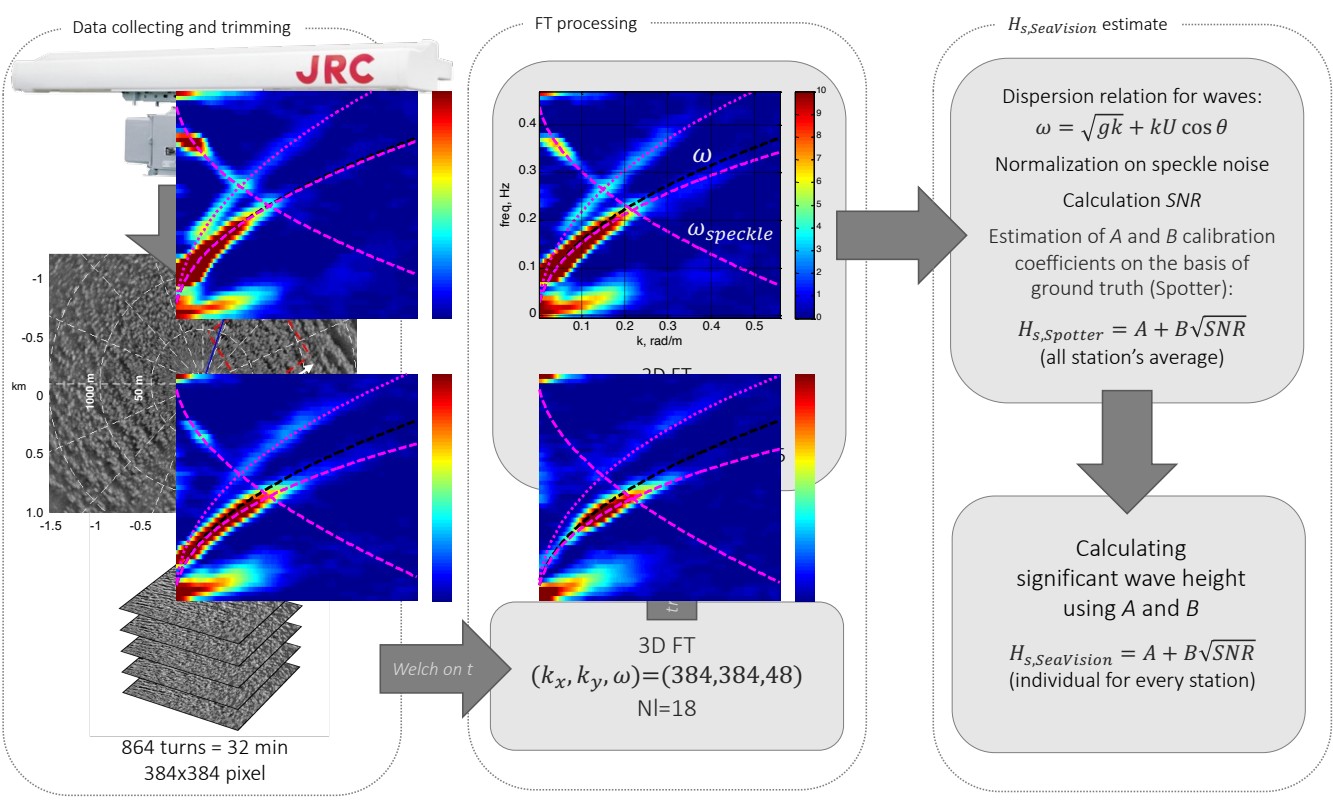

**Figure 3: Organigram of the data processing for estimation of wind waves parameters from the sea clutter images. JRC radar scanner (top left) is taken from the http://www.jrc.co.jp/eng/index.html.**

### 2.3 Spotter wave buoy data

To calibrate and validate SeaVision wave observations, we performed simultaneous measurements with the Spotter wave buoy (https://www.sofarocean.com/products/spotter) in the locations shown in Figure 1 and specified in Table A1. Once the ship is drifting at the location of the measurements, the Spotter buoy was deployed and started drifting away from the ship. Note that the ship drift is always faster compared with that of the buoy, thus, the distance between the buoy and the ship progressively increase. When the distance between the ship and the buoy reached at least 300 m, the "free floating" mode of SeaVision and Spotter buoy operation was initiated for at least 30 minutes as described in section 2.1. The longest free floating mode time period at some stations reached up to 1.5 hours. To ensure homogeneity of the analysis we used 20-minute segments from the "free floating" mode time series for further computations of significant wave height, wave spectra and directional moments: $H_s = 4\sqrt{E}$, where $E = \int_{0.01\,Hz}^{1.25\,Hz} E(f)df$ - the surface elevations variance in the frequency range of the wind waves. Further, we used wave parameters derived from the Spotter buoy as a "ground truth" for the calibration of

SeaVision data and derivation of A and B calibration coefficients in (2) (Table 2). Example of the wave energy spectrum for 20-minute Spotter buoy record is shown in Figure 4b.

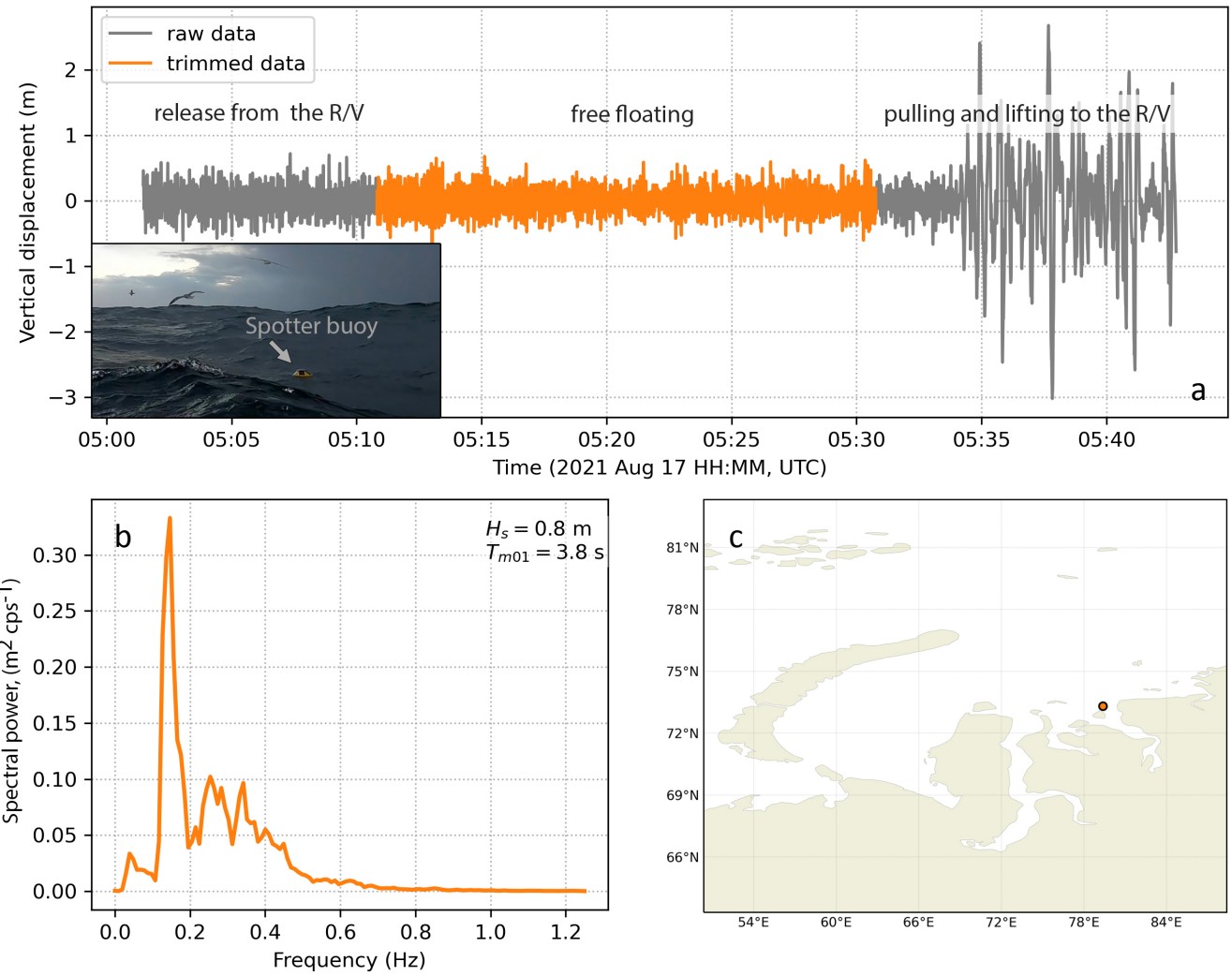

**Figure 4:** **Spotter wave buoy timeseries of vertical displacements at the station #3946 in the AI58 cruise (a), corresponding wave energy spectrum (b) and location of the station #3946 (c).**

**2.4 Meteorological data**

During all cruises, the AIRMAR WeatherStation 220WX was installed on the main ship mast at 30 m height above the sea. The weather station provided an output consisting of standard output parameters (barometric pressure, wind speed and

direction, air temperature and relative humidity). Wind characteristics were recalculated from the relative wind to the true wind in real-time mode.

275 **2.5 WaveWatch III model experiment**

We ran the WaveWatch III (WW3DG, version 6.07, WW3) spectral wave model forced by ERA5 reanalysis (Hersbach et al., 2020) over the domain and the time period of the research cruises (Table 3). The experiments were performed for the outer domain at 0.1° spatial and 1-hr temporal resolution and for the inner domain with 0.03° (~1 km) spatial (see Table 3)

280 and 1-hr temporal resolution. The outer domain solution was used for setting lateral boundary conditions for the inner domain. These experiments returned two-dimensional wave spectra co-located with SeaVision and Spotter buoy observations. In the WW3 experiments we used the ST6 parameterization (Bababin, 2006; Bababin, 2011; Rogers et al., 2012; Zieger et al., 2015) for wave energy input and dissipation and the discrete interaction approximation (DIA) scheme for nonlinear wave interactions (Hasselmann and Hasselmann 1985).

285

**Table 3: WW3 model configuration over the domains of expeditions.**

| Cruise | ASV50 | AI57 | AI58 |
|---|---|---|---|
| Region | North Atlantic polygon | North Atlantic polygon | Arctic polygon |
| Grid type | Regular, nested grid | Regular, nested grid | Curvilinear grid |
| Outer domain spatial resolution | 30°-75°N and 80W-10E 0.1°x0.1° | 30°-75°N and 80W-10E 0.1°x0.1° | 36°-90°N and 0°-360° 0.1°x0.1° |
| Inner domain spatial resolution | 54°-68°N and 45°W-1°E 0.03°x0.03° | 54°-68°N and 45°W-1°E 0.03°x0.03° | - |
| Time coverage | 2020.08.01-2020.09.06 | 2021.06.01-2021.07.12 | 2021.08.01-2021.09.30 |

**3 Results of validation of SeaVision measurements**

290

Validation of SeaVision data was provided for wind speeds from 2 to approximately 20 m/s and for significant wave heights from few tens of centimeters to 4.2 meters. Figure 5 demonstrates the results of the intercomparison of significant wave height ($H_s$) estimates retrieved from SeaVision data and those measured by the Spotter buoy and simulated with WaveWatch III. The $H_s$ differences 'Spotter minus SeaVision' (Figure 5a) and 'WW3 minus SeaVision' (Figure 5b) are plotted as a

295 function of wind speed recorded by the ship weather station (Table A1). Table 4 provides comparative estimates of differences in $H_s$ for the three cruises. On average WW3 yields lower wave heights than SeaVision $H_s$ by 28 cm, while the agreement between SeaVision and the Spotter buoy data is better with $H_s$ measured by Spotter being around 10 cm higher

than that retrieved from SeaVision. For low wind speeds SeaVision tends to underestimate $H_s$ up to 60 cm and for moderate and strong winds the analysis shows an overestimation of SeaVision $H_s$ compared to buoy and model data. This can be explained by better developed ripples (affecting the signal to noise ratio) at the ocean surface under stronger winds.

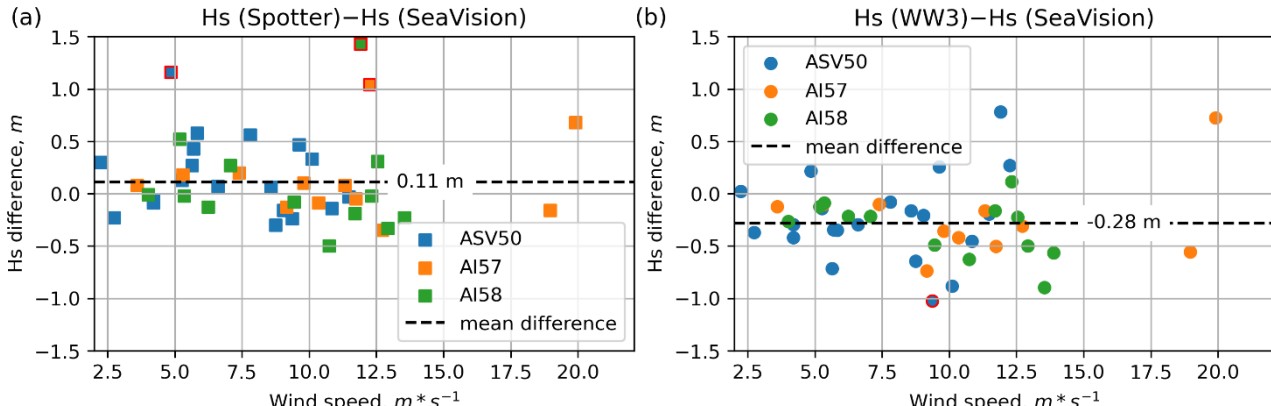

**Figure 5: Difference in the significant wave height ($H_s$) estimates for all stations as a function of the wind speed: Spotter buoy ("ground truth")** *minus* **SeaVision (a), WW3** *minus* **SeaVision (b). Dash lines mark the mean difference across all data points. Red squares and circles mark differences higher than 1 m.**

We also identified three locations (2901, 2928 and 2937, see Table A1) for which the differences between the Spotter buoy data and SeaVision reach more than 1 m (for 5 and 13 m/s winds). Weather conditions for these cases were not associated with severe weather and $H_s$ values were in the range between 1.5 and 2 m. However, in these cases we recorded a strong drift of the vessel due to the local current that potentially impacted the angle of the electromagnetic signal reflection from the surface and hence affected the accuracy of the radar images. Thus, strong ship drift may influence the SeaVision results and the data collected under strong ship drift should be considered with caution. These cases, in the future, can be identified by analysis of speed over ground (SOG parameter). For 'WW3 minus SeaVision' there is only one station #2841 where this difference reaches 1 m.

**Table 4: Differences in significant wave height estimates for the three cruises.**

| Mean difference in $H_s$ (m) | ASV50 | AI57 | AI58 |
|---|---|---|---|
| Spotter - SeaVision | 0.27 | 0.05 | -0.06 |
| WW3 - SeaVision | -0.24 | -0.24 | -0.36 |

Scatterplots for the $H_s$ and wave period ($T_{m01}$) demonstrate generally a better agreement between different data sources for $H_s$ (1.06 and 1.02 regression coefficients) than for $T_{m01}$ (1.05 and 0.86 regression coefficients) (Figure 6). There is no robust

evidence of the dependence of the magnitude or sign of $H_s$ and $T_{m01}$ differences on the magnitude of parameters themselves.

We also note that both SeaVision and Spotter show higher waves and slightly longer periods compared to WW3 (Figure 6). We note, however, that simulated wind waves with WW3 strongly depend on the atmospheric forcing (choice of reanalysis). Difference in climatological mean values over the North Atlantic obtained with WW3 but with different forcing functions can reach few tens of centimeters (Sharmar et al. 2021).

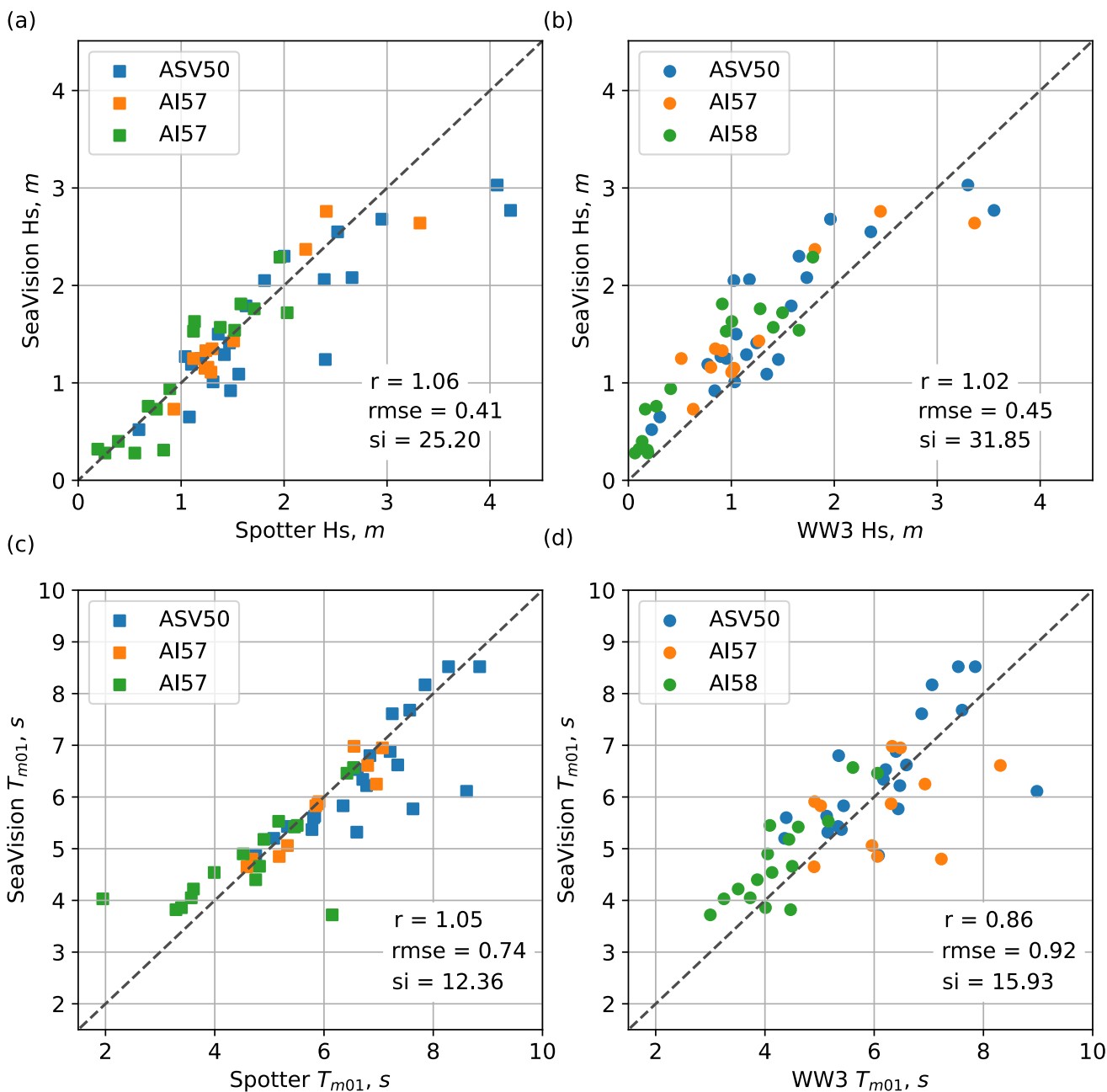

**Figure 6: Scatterplots of the significant wave height (H$_s$) and wave period (T$_{m01}$) revealed by SeaVision and measured by Spotter (a,c) as well as revealed by SeaVision and simulated with WaveWatch III (WW3, b,d) for all stations. Together with Root Mean Square Error (RMSE) and Scatter Index (SI) statistics.**

Overall, the analysis of significant wave heights among these three sources of data (Spotter, SeaVision and WW3) shows

that the highest H$_s$ values are measured by the Spotter buoy, lowest are simulated by WW3, with SeaVision being in between. These results are intuitively correct as wave buoys measure the actual elevations of ocean surface, SeaVision provides a proxy of local wave conditions from image analysis (thus imposing averaging over the domain) and is not expected to be as accurate as wave buoy data.

Figure 7 shows comparisons of wave directions (D$_s$) along with corresponding significant wave height (H$_s$) values

(simplified approximation of directional spectra) for six stations (see Table A1). Generally, all three data sources demonstrate very good agreement on directions (differences in waves direction do not exceed 10°) with corresponding wave height estimates being underestimated in model simulations as already mentioned above (Figures 5 and 6).

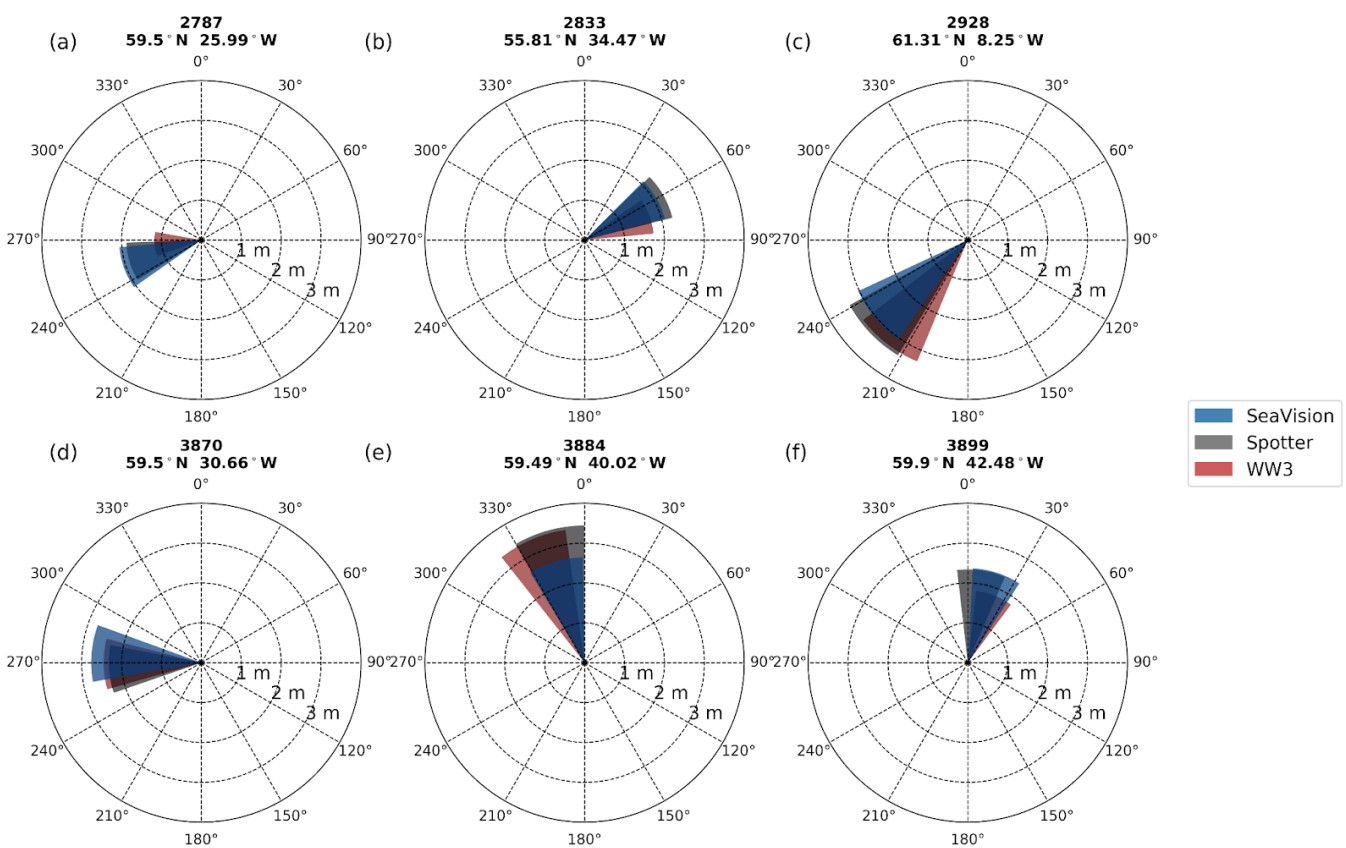


**Figure 7: Diagrams (roses) of mean wave direction (D$_s$, from) and significant wave height (H$_s$) on the basis of the three data sources: SeaVision (blue), Spotter (gray) and WaveWatch III (WW3, red) at the stations: #2787, #2833, #2928, #3870, #3884, #3899 (see Table A1).**

We also performed comparisons of SeaVision and Spotter H$_s$ estimates with satellite altimeter missions (Figures 8, 9). Figure 8 shows overpasses of all available satellite tracks of Jason-3, CFOSAT, Sentinel-3A, Sentinel-3B, SARAL and HaiYang-2B which are suitable for comparisons with our dataset. Altimeter data were used for comparisons when they satisfied two conditions: an overpass was within 2° latitude and within ±30 minutes from the measurement time (Table A1). In total we selected 20 cases that satisfied these conditions.

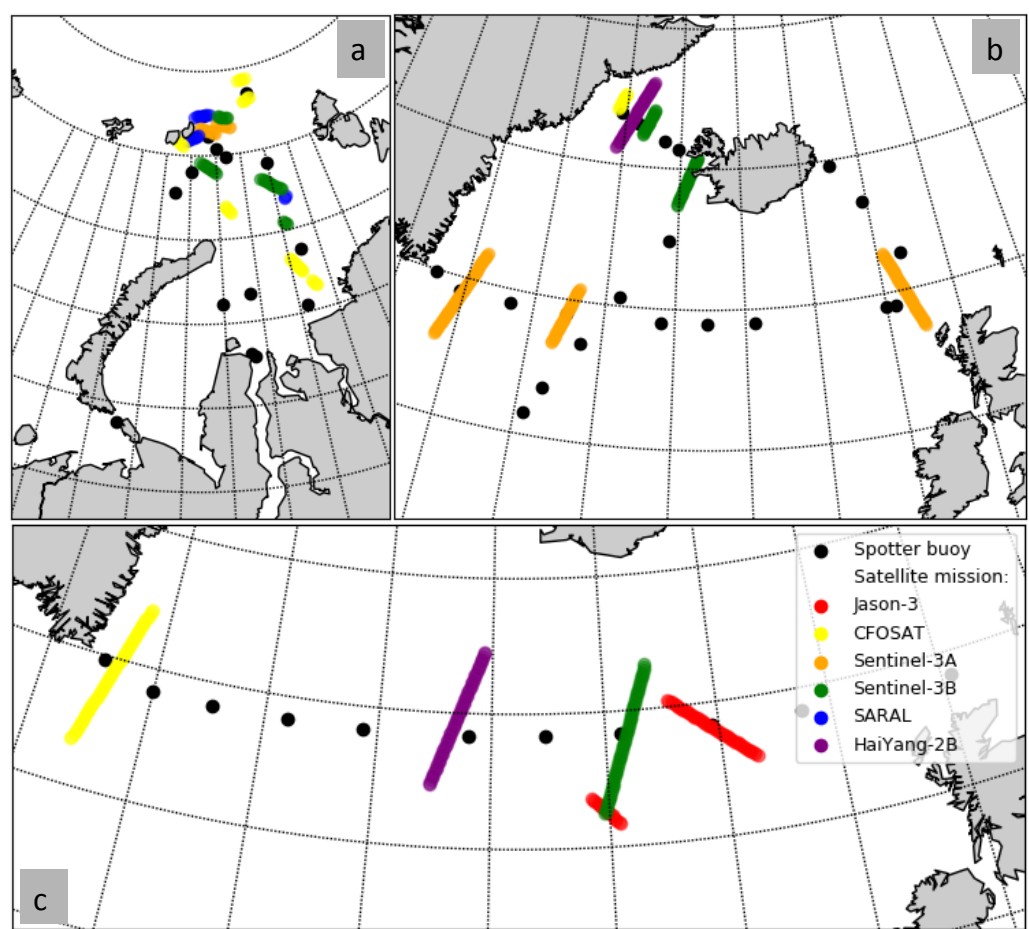


**Figure 8: Overpasses of satellite altimeter missions (Jason-3, CFOSAT, Sentinel-3A, Sentinel-3B, SARAL or HaiYang-2B) over the observational domains. Black dots indicate locations where wave parameters were measured simultaneously with Spotter wave buoy and SeaVision (Table A1).**

The average $H_s$ for these 20 locations measured by satellite altimeters is 1.47 m, with the Spotter buoy giving 1.38 m and SeaVision giving 1.26 m. There is a general agreement for most stations among these three sources of data and differences do not exceed 50 cm except for two cases: stations 2937 and 2901, where $H_s$ is underestimated by SeaVision comparing to Spotter and altimeter by more than 100 cm. These two outliers were already mentioned above (Figure 5) and large differences were attributed to a very strong drift of the ship for these locations.

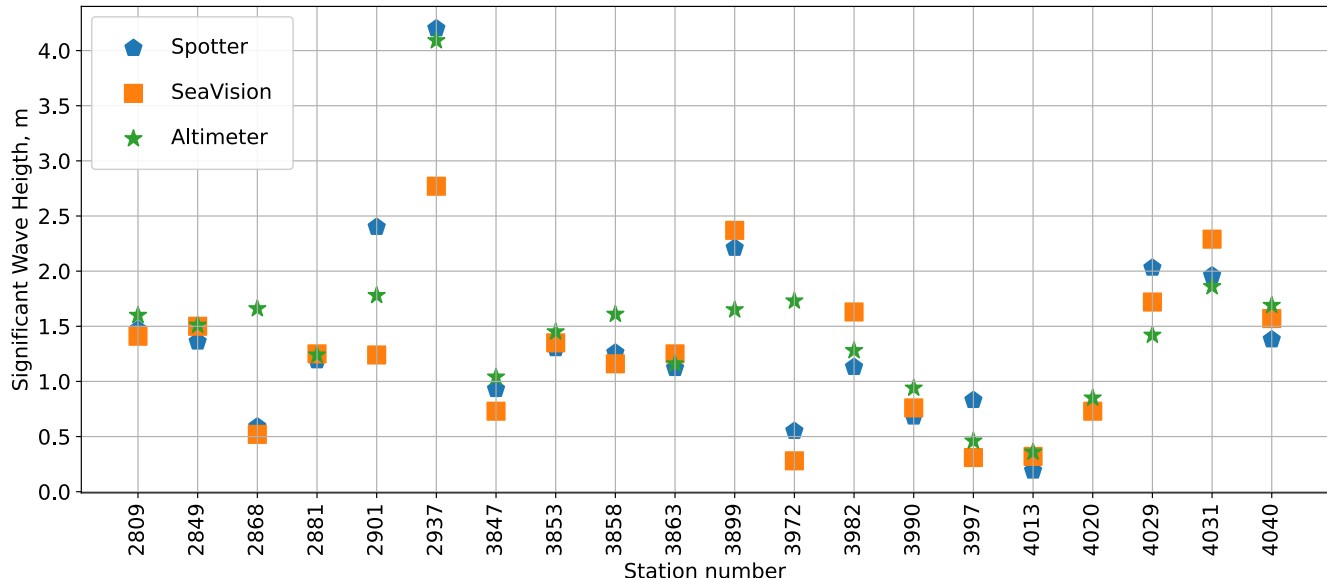


**Figure 9: Significant wave height estimates for the locations of satellite altimeter overpasses for three research cruises. Numbers on the horizontal axis correspond to the station numbering in Table A1.**

**Conclusions**


To broaden the avenue for widely needed broad-scale high-quality observations of ocean wind wave estimates we used a conventional navigation X-band ship radar equipped with a SeaVision recorder and software package. Here we present the evaluation of the instrument package for measuring wind wave parameters and comparing them with in-situ observations and model results. The data were collected on three cruises in the subpolar North Atlantic and in the Kara Sea. All SeaVision

records were co-located with in-situ Spotter buoy measurements, which were used for validation. We demonstrate an overall agreement of the estimates of significant wave height and wave period measured by SeaVision with the Spotter buoy measurements and with simulations using the WW3 spectral wave model. Estimates of significant wave height between SeaVision, WW3 and the Spotter buoy are in a better agreement than those for the wave periods. In the ranges of $H_s$ up to 4.2 m the average difference between the Spotter buoy and SeaVision is around 10 cm for $H_s$, while WW3 simulations are

lower than SeaVision $H_s$ by 28 cm. We note, however, that comparisons with WW3 should be considered with caution, as the model results are significantly dependent on the choice of forcing function (atmospheric reanalysis). SeaVision tends to

underestimate mean wave periods by ~0.5 s compared to the Spotter buoy while the differences in periods with WW3 simulations may amount to more than 2 sec. Also, a very good agreement was found for the wave directions whose spread across all three data sources does not exceed 10°.

We present the newly developed SeaVision system for digitizing and recording the analog signals from navigation radars, and further providing quantitative estimates of wind wave characteristics. A broad implementation of SeaVision opens a potential for enhancing massive observations of wind waves over the open ocean. SeaVision is currently mounted onboard two R/Vs operated by IORAS, but in 2022 five more IORAS R/Vs will be supplied with SeaVision systems. Data records will become operationally available at an open source web page. In 2023 we also plan to develop a portable and cheaper

version of SeaVision that can be easily mounted onboard of any commercial ship with navigational radar operating in the open ocean, as well as on the platform, lighthouse or any coastal infrastructure. After further validation in different sea state and weather conditions we plan to upgrade SeaVision to portable device and to incorporate all post-processing procedures into the internal software package that will make it possible for commercial ships on which the system is installed to provide real-time reporting of wind wave parameters though the Global Telecommunication System (GTS). Theoretically the

estimated data flow is formally one estimate per 2-3 seconds (1 full turn of the radar antenna). Even with a reporting frequency once per minute, the potential of the SeaVision data flow exceeds the current VOS data flow by hundred times. Contrasting to existing commercial systems for wind waves monitoring with navigational marine radars, such as WaMoS II (http://www.oceanwaves.de), SeaDarQ (http://www.seadarq.com/) and WaveFinder (Park et al., 2006), SeaVision is representing potentially a low-cost, portable and easy to install alternative. Wide use of such a system on commercial ships

can drastically increase the amount of sea state observations available to users, including the National Meteorological Offices using this information as data assimilation input for NWP models and reanalyses.

The Global Climate Observing System (GCOS) and associated Global Ocean Observing System (GOOS) are considering the sea state to be a critical climate variable highly demanded by global observing modules. We hope that SeaVision with its perspective to provide exceptionally high global coverage with on-line wave measurements will meet this urgent demand

and help to satisfy GCOS given its mandate for systematic observations under the UN Framework Convention on Climate Change (UNFCCC), including also GCOS and GOOS responsibilities under the Subsidiary Body for Scientific and Technological Advice (SBSTA) and the Subsidiary Body for Implementation (SBI).

**Data availability**


Datasets that contains significant wave heights, wave periods, wave directions, wave energy frequency spectra, meteorological data and other related parameters from both SeaVision and the Spotter buoy at the locations of every station (Table A1) is available through the PANGAEA repository - https://doi.org/10.1594/PANGAEA.939620 (Gavrikov et al., 2021). In this dataset we provide wind waves statistics disregarding separation of the swell and wind waves at this stage of

the SeaVision development. We plan to include this procedure into the next studies. At the same time, we provide one

dimensional spectrum that potentially allows to see first and seconds peaks associated with winds waves and swell (an example is shown in Figure 4b). Users interested in the analysis of the raw radar dataset or in the wave characteristics in the locations were measurements were carried out only with SeaVision are welcome to request access from Alexander Gavrikov (gavr@sail.msk.ru).


**Appendix A: List of the locations (stations) of the wind waves measurements during three research cruises**

**Table A1: Stations list: geographical locations and time of all stations where the wind waves measurements were performed simultaneously with SeaVision and Spotter buoy. In the last column letters stand for the name of the research vessel (ASV – "Akademik Sergey Vavilov", AI – "Akademik Ioffe") and numbers stand for the sequence**
**number of reseach cruise since the beginnig of the reseach vessel operation.**

| # | Station # | Start UTC time | End UTC time | Latitude° N | Longitude° E | Cruise # |
|---|-----------|----------------|--------------|-------------|--------------|----------|
| 1 | 2868 | 27.08.2020 13:53 | 27.08.2020 14:13 | 65.67 | -25.26 | ASV50 |
| 2 | 2881 | 28.08.2020 10:45 | 28.08.2020 11:05 | 66.49 | -28.89 | ASV50 |
| 3 | 2885 | 28.08.2020 19:05 | 28.08.2020 19:25 | 66.84 | -30.43 | ASV50 |
| 4 | 2763 | 11.08.2020 11:25 | 11.08.2020 11:45 | 59.50 | -10.00 | ASV50 |
| 5 | 2777 | 13.08.2020 18:15 | 13.08.2020 18:35 | 59.50 | -19.32 | ASV50 |
| 6 | 2782 | 14.08.2020 18:42 | 14.08.2020 19:02 | 59.50 | -22.66 | ASV50 |
| 7 | 2787 | 15.08.2020 18:10 | 15.08.2020 18:30 | 59.50 | -25.99 | ASV50 |
| 8 | 2797 | 17.08.2020 10:12 | 17.08.2020 10:32 | 59.50 | -32.67 | ASV50 |
| 9 | 2803 | 18.08.2020 12:17 | 18.08.2020 12:37 | 59.50 | -36.67 | ASV50 |

| | | | | | | |
|---|---|---|---|---|---|---|
| 10 | **2809** | 19.08.2020 13:26 | 19.08.2020 13:46 | 59.50 | -40.34 | ASV50 |
| 11 | **2821** | 20.08.2020 13:44 | 20.08.2020 14:04 | 59.90 | -42.32 | ASV50 |
| 12 | **2833** | 22.08.2020 15:26 | 22.08.2020 15:46 | 55.81 | -34.47 | ASV50 |
| 13 | **2841** | 23.08.2020 12:31 | 23.08.2020 12:51 | 56.78 | -33.53 | ASV50 |
| 14 | **2849** | 24.08.2020 14:06 | 24.08.2020 14:26 | 58.53 | -31.43 | ASV50 |
| 15 | **2856** | 25.08.2020 12:42 | 25.08.2020 13:02 | 60.30 | -29.04 | ASV50 |
| 16 | **2863** | 26.08.2020 11:45 | 26.08.2020 12:05 | 62.40 | -25.73 | ASV50 |
| 17 | **2901** | 30.08.2020 13:05 | 30.08.2020 13:25 | 65.94 | -26.49 | ASV50 |
| 18 | **2903** | 01.09.2020 13:05 | 01.09.2020 13:25 | 64.82 | -12.49 | ASV50 |
| 19 | **2913** | 02.09.2020 10:17 | 02.09.2020 10:37 | 63.35 | -10.38 | ASV50 |
| 20 | **2928** | 03.09.2020 19:24 | 03.09.2020 19:44 | 61.31 | -8.25 | ASV50 |
| 21 | **2937** | 04.09.2020 21:16 | 04.09.2020 21:36 | 59.50 | -9.31 | ASV50 |
| 22 | **3831** | 29.06.2021 19:49 | 29.06.2021 20:09 | 59.50 | -4.60 | AI57 |
| 23 | **3841** | 01.07.2021 09:26 | 01.07.2021 09:46 | 59.49 | -11.33 | AI57 |
| 24 | **3847** | 02.07.2021 10:33 | 02.07.2021 10:53 | 59.50 | -15.33 | AI57 |

| 25 | 3853 | 03.07.2021 12:35 | 03.07.2021 12:55 | 59.50 | -19.33 | AI57 |
|----|------|------------------|------------------|-------|--------|------|
| 26 | 3858 | 04.07.2021 11:38 | 04.07.2021 11:58 | 59.50 | -22.67 | AI57 |
| 27 | 3863 | 05.07.2021 10:05 | 05.07.2021 10:25 | 59.50 | -26.00 | AI57 |
| 28 | 3870 | 06.07.2021 16:29 | 06.07.2021 16:49 | 59.50 | -30.67 | AI57 |
| 29 | 3875 | 07.07.2021 15:57 | 07.07.2021 16:17 | 59.52 | -33.98 | AI57 |
| 30 | 3880 | 08.07.2021 17:32 | 08.07.2021 17:52 | 59.50 | -37.33 | AI57 |
| 31 | 3884 | 09.07.2021 13:51 | 09.07.2021 14:11 | 59.50 | -40.00 | AI57 |
| 32 | 3899 | 11.07.2021 12:45 | 11.07.2021 13:05 | 59.90 | -42.48 | AI57 |
| 33 | 3911 | 12.08.2021 13:27 | 12.08.2021 13:47 | 70.37 | 58.04 | AI58 |
| 34 | 3929 | 14.08.2021 21:43 | 14.08.2021 22:03 | 75.15 | 75.09 | AI58 |
| 35 | 3930 | 15.08.2021 06:40 | 15.08.2021 07:00 | 73.98 | 72.66 | AI58 |
| 36 | 3939 | 16.08.2021 12:40 | 16.08.2021 13:00 | 73.75 | 73.66 | AI58 |
| 37 | 3946 | 17.08.2021 04:53 | 17.08.2021 05:13 | 73.31 | 79.35 | AI58 |
| 38 | 3956 | 18.08.2021 12:52 | 18.08.2021 13:12 | 75.14 | 79.54 | AI58 |
| 39 | 3972 | 21.08.2021 12:27 | 21.08.2021 12:47 | 82.14 | 78.88 | AI58 |

| 40 | 3982 | 22.08.2021 15:48 | 22.08.2021 16:08 | 81.93 | 73.70 | AI58 |
|---|---|---|---|---|---|---|
| 41 | 3990 | 23.08.2021 14:43 | 23.08.2021 15:03 | 81.44 | 67.25 | AI58 |
| 42 | 3997 | 24.08.2021 08:02 | 24.08.2021 08:22 | 81.04 | 72.66 | AI58 |
| 43 | 4013 | 25.08.2021 19:28 | 25.08.2021 19:48 | 79.93 | 72.11 | AI58 |
| 44 | 4020 | 26.08.2021 13:20 | 26.08.2021 13:40 | 79.51 | 65.06 | AI58 |
| 45 | 4025 | 27.08.2021 03:05 | 27.08.2021 03:25 | 78.28 | 65.33 | AI58 |
| 46 | 4029 | 27.08.2021 12:39 | 27.08.2021 12:59 | 77.67 | 65.45 | AI58 |
| 47 | 4031 | 27.08.2021 18:30 | 27.08.2021 18:50 | 77.86 | 64.85 | AI58 |
| 48 | 4040 | 28.08.2021 11:11 | 28.08.2021 11:31 | 78.84 | 61.62 | AI58 |

**Appendix B: Methodology for the computation of wave parameters from sea clutter images**

We stated above (section 2.2.2) that to relate the signal to the wind waves, we assume that components of the spectrum outside of the dispersion relation are related to the background speckle-noise and components of the spectrum that satisfy the dispersion relation (1) related to the signal, associated with the wind waves. Eq. (1) presents the dispersion relation for the first harmonic and can be also easily extended to the second harmonic as follows:

$$\omega_{n,2}(k) = \sqrt{2gk} + 2k \cdot U \cdot cos\,\theta_n \tag{B.1}$$

Here and later index $n$ refers no the number of directional sector (22.5º width each). The curve associated with the second harmonic is clearly seen in Figure 3. The rest of the signal lying in the spectral domain outside the bands associated with dispersion curves and attributed to speckle-noise (Kanevsky, 2009) is needed to be properly quantified. This depends on the

algorithm used for the quantification of bands associated with dispersion relation curves. Speckle-noise is used for

normalization of the radar spectrum and removing the impulse power impact on the radar signal modulations by the sea waves (Kanevsky, 2009). The 2D normalized spectrum $S_{n,2d,norm}(k,f)$ of the signal at each wavenumber k can be calculated as:

$$S_{n,2d,norm}(k,f) = \frac{S_{n,2d,image}(k,f)}{\int S_{n,2d,image,\omega\, speckle}(k,f)df} - 1 \tag{B.2}$$

where the speckle frequency is:

$$\omega_{speckle} = (f \notin \omega_{n,1}/2\pi \ and \ f \notin \omega_{n,2}/2\pi) \tag{B.3}$$

Then the full image spectrum $S_{n,1,\omega}(f)$ needs to be filtered to obtain the power corresponding to the band capturing the first $\omega_{n,1}$ harmonic for the direction n:

$$S_{n,1,\omega}(f) = \int_{k_{n,1}-\Delta k}^{k_{n,1}+\Delta k} S_{n,2d,norm}(k,f)dk \tag{B.4}$$

Here $k_{n,1}$ is the dispersion relation (1) solution for the first harmonic $\omega_{n,1}(k_{n,1}) = 2\pi f$ and $\Delta k$ is related to the size of the processing area (720 m) as $\Delta k = 0.02 \approx 2 \cdot 2\pi/720$ (rad/m). Similarly for the second harmonic $\omega_{n,2}$ we obtain:

$$S_{n,2,\omega}(f) = \int_{k_{n,2}-\Delta k}^{k_{n,2}+\Delta k} S_{n,2d,norm}(k,f)dk \tag{B.5}$$

where $k_{n,2}$ is the dispersion relation (B1) solution for the second harmonic $\omega_{n,2}(k_{n,2}) = 2\pi f$.

The total power $S_{n,\omega}(f)$ falling in the bands along dispersion relation curves yields:

$$S_{n,\omega}(f) = S_{n,1,\omega}(f) + S_{n,2,\omega}(f) \tag{B.6}$$

Given that this procedure is applied to all 16 sectors of the image (see section 2.2.2), the omnidirectional image frequency spectrum $S_{image}(f)$ can be derived as follows:

$$S_{image}(f) = \frac{1}{16}\sum_{n=1}^{16} S_{n,\omega}(f) \tag{B.7}$$

Further integration over the frequency domain returns the zeroth moment $m_{0,image}$ of the $S_{image}(f)$ spectrum:

$$m_{0,image} = \int_{fr^8}^{fr^{48}} S_{image}(f)df, \tag{B.8}$$

which provides us with the estimate of *SNR* being:

$$SNR \equiv m_{0,image} + 1.$$

The limits of the integration in (B.8) are $fr^8 = 8\Delta f$ and $fr^{48} = 48\Delta f$, where $\Delta f = \frac{rpm}{60 \cdot 48}$ is defined by the antenna rotation speed $rpm$ (rotations per minute, Table 2) and the 48-points size window of FT in the time domain.

Formally, considering the $S_{image}(f)$ spectrum as a modulation analog of the real sea wave spectrum, $S_w(f)$, the zeroth moment $m_{0,image}$ can be further converted to the magnitude of signal modulations $H_{image}$ on the radar image which stands as a provisional measure of $H_s$:

$$H_{image} = 4\sqrt{m_{0,image}} \tag{B.9}$$

Further the transform of the omnidirectional SeaVision image frequency spectrum $S_{image}(f)$ to the sea wave frequency (wave energy) spectrum $S_{w,SeaVision}(f)$ visible by SeaVision is performed by applying the standard technique described in section 2.2.2 and resulting in Eq. (2) returning significant wave height $H_{s,SeaVision}$ estimate based on the radar calibration coefficients A and B along with estimate for the wind wave period (3) derived from the zeroth and the first moments of the spectrum.

The mean wave direction $D_{s,SeaVision}$ is estimated with the centroid method

$$D_{s,SeaVision} = \frac{180}{\pi} arg\left(\frac{\sum_{n=1}^{16} exp\left(i\frac{\pi}{180}\theta_n\right)D_n}{\sum_{n=1}^{16} D_n}\right), \tag{B.10}$$

$$D_n = \int_{fr8}^{fr48} S_{n,\omega}(f)\,df, \tag{B.11}$$

where $arg$ is the argument of the complex number, $D_n$ is the zeroth moment of the spectrum $S_{n,\omega}(f)$ in the direction $n$.


**Appendix C: Definition of all the parameters in the manuscript and dataset.**

| Parameters | Short name | Definition | Range | Name in netcdf |
|---|---|---|---|---|
| Meteorological variables | | | | |
| Wind Speed ($m \cdot s^{-1}$) | $U_{10}$ | - | - | meteo_wspd |
| Wind Direction (°) | $\theta_{10}$ | - | $0 - 360°$ (from) | meteo_wdir |
| Atmospheric Pressure (hPa) | $p$ | - | - | meteo_pres |
| Atmospheric Temperature (°C) | $T$ | - | - | meteo_temp |
| Humidity (%) | $H$ | - | $0 - 100\ \%$ | meteo_humd |
| Spotter wave buoy variables | | | | |
| 1-D wave energy spectrum ($m^2 \cdot Hz^{-1}$) | $S_{w,Spotter}$ | $\int_{f4}^{f127} S(f)df$ | $0.01 - 1.25$ Hz | buoy_Szz, buoy_freq |
| Significant Wave Height (m) | $H_{s,Spotter}$ | $4\sqrt{m_{0,Spotter}}$ | $0.2 - 4.2$ m | buoy_hs |
| Energy Wave Period (s) | $T_{m01,Spotter}$ | $\dfrac{m_{0,Spotter}}{m_{1,Spotter}}$ | $1.85 - 8.85$ s | buoy_ts |

| Mean Wave Direction (°) | $D_{s,Spotter}$ | $270^0 - \dfrac{180^0}{\pi} arctan\, 2(b_1, a_1)$ | $0 - 360°$ (from) | buoy_ds |
|---|---|---|---|---|
| SeaVision variables | | | | |
| 1-D wave energy spectrum ($m^2 \cdot Hz^{-1}$) | $S_{w,SeaVision}$ | Eq. 6 | 0.0423 – 0.4069 Hz | radar_Szz radar_freq |
| Significant Wave Height (m) | $H_{s,SeaVision}$ | $4\sqrt{m_{0,SeaVision}}$ | 0.3 – 3 m | radar_hs |
| Energy Wave Period (s) | $T_{m01,SeaVision}$ | $\dfrac{m_{0,SeaVision}}{m_{1,SeaVision}}$ | 3.7 – 8.5 s | radar_ts |
| Mean Wave Direction (°) | $D_{s,SeaVision}$ | Eq. B.10 | $0 - 360°$ (from) | radar_ds |

**Author contributions**

NT, AG, DI, VS, AS, LS, VS and PS participated in research cruises and data collection. The leading role in field work program development and implementation belongs to AG and VS. DI made preprocessing, postprocessing and analysis of the radar (SeaVision) dataset, VS developed the configuration and ran the WW3 model for the period of cruises, AG analysed all Spotter buoy data. EE, AG and VS did validation with satellite missions. VF, BT and SB provided hardware development and mounting of the SeaVision to the research vessels. VT, OR and AS provided operational support for the research cruises. NT has a leading role in the project set up and manuscript writing. The initial idea of the research belongs to SG. The scoping of the manuscript was developed by NT, SG and KPK. All authors contributed to the discussion, interpretation of the results and writing.

**Competing interests**

The authors declare no conflict of interests.

**Acknowledgements**

We thank Editor Dr. Giuseppe M.R. Manzella, anonymous Reviewer 1 and Dr. Alamgir Hossan (Reviewer 2) for careful examination of the manuscript and useful comments and suggestions that allowed to improve the manuscript significantly. We also thank Prof. Ian Young and Dr. Vladimir Karaev for their comments during the open discussion stage. We acknowledge Dr. Igor Skvortsov and the Atlantic branch of the Shirshov Institute of Oceanology in Kaliningrad for the local support and the crews of research vessels "*Akademik Sergey Vavilov*" and "*Akademik Ioffe*" for their help in setting up buoy measurements in the open ocean. Authors thank the "Floating University" scientific and educational program, Dr. Natalia

Stepanova, Dr. Alexander Osadchiev, Dr. Sergey Gladyshev and Dr. Vsevolod Gladyshev of Shirshov Institute of Oceanology for their leading roles in organisation of research cruises. We are also thankful to Dr. Vika Grigorieva and Dr. Igor Goncharenko of Shirshov Institute of Oceanology for their help with setting up the hardware for radar signal digitization and useful advice. We thank Dr. Takaya Uchida at Université Grenoble Alpes for final proofreading of the manuscript.

**Financial support**

This study was funded by the Russian Foundation for Basic Research, project № 20-35-70025. VS and AG were also supported with grant № 17-77-20112-P from the Russian Science Foundation (WW3 setting for the period of the research cruises). SG was supported by the Ministry of Science and Higher Education of the Russian Federation (agreement 075-15-2021-577, interpretation and analysis of observational biases).

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
