# Peer review of "Wind waves in the North Atlantic from ship navigational radar: SeaVision development and its validation with Spotter wave buoy and WaveWatch III"

_Earth System Science Data, 2021_

## Referee Comment (RC1)

**Specific comments**

Abstract:
"Measurements with SeaVision were quality controlled and validated by comparison with Spotter buoy data and WaveWatch III experiments."
One of my main criticisms to this work is the use of wave modelling for validation of in-situ observations. It needs to be considered that numerical models provide an approximation of physical phenomena, and they need to be quality controlled against observations, it does not sound logical/scientific to go the other way around.

Section 2.1:
I suggest more elaboration of the expeditions in this section considering that the main purpose of this manuscripts is to present the dataset which has been collected during the expeditions. For example, information such as region, location (including approximate coordinates), date (including day of month), name of the vessel, distance sailed, start and end point, stations, instruments used, metocean parameters collected, general depth etc. should be included in the text.

Figure 1:
Explain what the reason is behind measurement gaps as depicted in panels (a) and (c).
Briefly explain cruise numbers (50, 57, 58) in the text of the section. For instance, cruise 50 out of how many cruises per month/year aboard RV Academic Survey Vavilov?
Indicate what RV stand for. For instance, by adding "research vessel (RV)" in the caption.
The published dataset at https://sail.ocean.ru/tilinina2021/ seems to be including the measurements at stations shown with orange dots only (Spotter and SeaVision). I wonder why the rest of the observations (SeaVision only measurements shown in green dots) have not been published?
Further, the number of stations shown orange dots are not consistent between this figure, Appendix A, and the published dataset. In total, 52 points are shown Figure 1. However, 50 stations are provided in Appendix A and observations at 48 stations have been published. From Cruise 57, station number 3836 is not included in the dataset but provided in Appendix A. From cruise 50, stations 2771 and 2792 are included in the Appendix but not in the dataset and station 2913 is in the dataset but not in the Appendix.

Line 115: "other regular deep ocean observations"
What are these "other regular deep ocean observations"? Elaborate more or provide examples.

Lines 116 to 119: "In particular, two cruises in the North Atlantic (Figure 1a,b) are related to regular deep ocean observations at the 59,5°N (Verezemskaya et al., 2021; Falina et al. 2007; Gladyshev et al. 2018, 2019; Sarafanov et al. 2008, 2018) and the Arctic expedition is a part of the IO RAS "Floating University of IORAS" program (Stepanova, 2018)."
This sentence is confusing. It is providing information of the first two expeditions versus the "program" which the third expedition is associated with. Also, it has already been mentioned at lines 115-116 that all three expeditions are carried out by IO RAS. So, what is the need to repeat it here about the "Arctic expedition"?

Line 118: "Arctic expedition is a part of the IO RAS"
What is "Arctic expedition" referring to? The expeditions have not been named previously.

Line 121: "During all "stations""
It is not clear what the word "stations" is referring to i.e,, no information about any station has been previously defined in the text.

Lines 122 and 123: "to provide conditions for Spotter buoy wave observations in the free floating mode."
Is "free floating" referring to the deployment of spotter buoy? If yes, it needs to be clarified in the text. Also, explain why this deployment option has been targeted.

Section 2.2:
I think this section can be improved by being organised in a more informative manner.
I was expecting a thorough description of SeaVision system and its comparison and superiority against other widely used observational systems such as WaMoS-II as it is indicated in the conclusion that "the commercial oceanographic systems for the wind waves monitoring such as WaMoS II, SeaDarQ and WaveFinder already exist … [this manuscript's] main aim is to develop in the nearest future a low cost, independently operating and stable system that would allow to broaden observational network for the wind waves".
Explain why the dataset only includes one dimensional frequency wave spectrum and no information is provided about the directionality of waves.
Underway observations and their file types deserve a much more comprehensive explanation considering the scope of ESSD journal.

Line 133: "For our purposes we used the shortest possible pulse length"
Explain what the reason is for setting the radar to short pulse length i.e., how the resolution of images may be affected etc.

Line 162: "this results into the three-dimensional spectrum S(kx,ky,f)"
Explain what Kx and Ky are.

Line 164 and 165: The correct expression of the linear dispersion relation is: $\omega = \sqrt{gk \tanh (kh)}$. The angular frequency has been previously defined as $\omega$, why has it been changed to $\Omega$ in this equation? The unit of g (gravitational acceleration) should be corrected to m*s$^{-2}$ and H should be replaced with h which is the depth, not wave height!! Also, correct the equation in Figure 3.

Line 166: "the signal ($\Omega_{BGN}$),"
What is this signal? What does "BGN" stand for?

Line 185 and 186: "wave period Ts estimated traditionally using the first moment of the spectrum."
Provide the equation for the wave period calculations. I also suggest using the standard annotation of wave period based on the first moment "$T_{m1}$" or "$T_{m01}$" instead of "$T_s$".

Line 191: "Table A1 provides a list of all locations where SeaVision+Spotter buoy (or SeaVision only) measurements were carried out."
From the published dataset, I can see in all the stations provided in Table A1 both SeaVision and Spotter data exist. No data is available at stations with SeaVision only measurements.

Lines 200 and 201: "Analysis of the raw vertical and horizontal displacements recorded by buoy starts from the selecting in timeseries the "free floating" measurements"
Explain in the text how free-floating measurements have been identified from timeseries.

Line 202: "using common definitions (see Appendix in Raghukumar et al., 2019)"
I suggest adding an appendix that provides definition of all the parameters in the manuscript/published dataset.

Line 203: "in the frequency range of the wind waves."
I wonder why you have limited yourself to wind waves only, both from the spotter buoy in this section as well as SeaVision and WaveWatch-III in other sections of the manuscript. What is

the reason behind disregarding swell observations? Couldn't it be included in the dataset/manuscript? If not, it needs to be clarified why.

Further, a range of 0.05 to 0.3 Hz seems to be underestimating frequency of wind waves. For example, from Semedo et al., (2009) frequency of wind sea in North Atlantic seems to be above 0.2 to 0.3 Hz in summer when IORAS expeditions have taken place (see Figure 7 at Semedo et al., (2009)). The frequency/ period ranges can also be confirmed from example wave spectrum in Figure 4 of current manuscript.

Please also confirm the frequency range applied to the published data (one dimensional frequency spectra) for calculation of integrated wave parameters provided in the "Global Attributes" of the NetCDF files i.e., Hs_radar , Ts_radar, Hs_buoy, Ts_buoy.

Figure 4:
It should be indicated that this spectrum is an example. Please also provide the station number where this spectrum has been recorded.

From the published dataset, I can see that only one-dimensional frequency wave spectra are provided. I am surprised to see the information about wave directions in the bottom right panel. Is there a point I am missing here?

It may be better to plot the wave spectrum in frequencies instead of periods.

Indicate what PSD stands for. Also, describe what $\theta_p$ and $\sigma_{\theta p}$ are.

I suggest labelling different panels with (a), (b), (c), etc. This also applied to Figures 2 and 3.

Section 2.4:
I cannot find any information regarding the meteorological data in the published dataset at https://sail.ocean.ru/tilinina2021/. It would be beneficial to publish the meteorological data (after being quality controlled) together with the wave observations.

Section 2.5:
A WaveWatch-III model for this study need a much more extensive explanation than one paragraph only. Model physics and packages, setup, calibration, and validation should be comprehensively explained. I am surprised to see that no information is provided about model calibration and validation.

Lines 223 to 225: "We run WaveWatch III (WW3DG, version 6.07, WW3) spectral wave model with ERA5 reanalysis (Hersbach et al., 2020) as lateral boundary conditions with 0.1° spatial and 1 hourly temporal resolutions."

What does it mean to run the model with ERA5? The model's forcing and boundary conditions (parameters with their temporal and spatial resolution) need to be clearly explained. It is not clear whether "0.1° spatial and 1 hourly temporal resolutions" refers to the resolution of forcing and boundary conditions or the actual wave model.

Line 238 to 240: "In general, for the lower wind speeds SeaVision underestimates Hs by up to 50 cm and overestimates Hs for the higher wind speeds. This effect can be due to better ripples development on the ocean surface during higher winds affecting the signal to noise ratio (Formula 1)."

I am a bit confused here, from Figure 5(a) the difference between Hs observed with Spotter and SeaVision are less, and it increases for wind speeds more that 8 m/s. The text seems to be indicating the opposite while describing better ripple developments at high winds.

Line 247: "there are two stations (2901 and 2937 see Table A1) where this difference reaches almost 100 cm"

Show these points in Figure 5 by a label and/or different colour.

Figure 6 (and lines 253 to 259):
The fitted lines should cross the [0 0] points i.e., logically when Spotter record a wave height (or wave period) equal to zero, SeaVision should return zero, too (it is similarly true about WaveWatch-III estimates).
Overestimation of wave height and period by WaveWatch-III against Spotter measurements may be indicating that the model has not been properly calibrated.
I suggest including other error statistics such as Root Mean Square Error (RMSE) and Scatter Index (SI).
I wonder why the number of points in these plots (~32) is less than the number of stations (~50)? Surprisingly, I get different looking plots from the published dataset while reading the values from "Global Attributes" of each NetCDF file published. Below is the significant wave height from buoy versus radar, for example:

[Figure]

Lines 265 and 266: "waves direction (from)"
Does it mean the wave directions are in "coming from" convention? It needs to be further explained.

Figure 7:
This figure needs a more extensive explanation. The sources of errors and inconsistencies need to be described.
Similar to figure 4, I am surprised to see directional wave roses while no information about directionality of waves in provided in the published dataset, please explain.

Data Availability:
I suggest including a calendar as well as a metadata in the repository to provide information about the dataset each folder/link includes.
Parameters/information available in the dataset and their description (including an extensive description of Global Attributes and Variables in NetCDF files) are worth being added to the

manuscript, probably as an Appendix. Also, indicate the full length of time that the measurements are covering.

**Technical corrections**

The English language, specifically grammar and punctuations, need to be revised and corrected throughout the text.

Consistent symbols and annotations should be used in the text and figures. For example, the WaveWatch-III wave model is indicated by "WaveWatch III" in some parts and "WW3" in others. Other examples are:
Significant wave height being indicated by "H" and "Hs", and
Figure being indicated by "Fig." and "Figure".

Line 135: "SeaVision system (Fig. 2) connected to the radar via splitter, **it** digitizes and records directionally"
Replace with "SeaVision system (Fig. 2)**,** connected to the radar via splitter, digitizes and records directionally".

Line 154: "FFT-based"
It should be indicated what FFT stands for. Replace with "Fast Fourier Transform (FFT) based".

Line 175: "to estimate**s to"**
Replace with "to estimate".

Line 176: "spectra power"
Replace with "spectra**l** power".

Line 121: "During all "stations""
Replace with "**At** all stations".

Line 190: "waves observations"

Replace with "wave observations".

Line 199: "we use"
Replace with "we used".

Line 279: "Nowadays there is still exists gap"
Correct the sentence grammatically.

Line 279: "winds waves is"
Replace with "wind waves **are**".

Line 280: "component"
Replace with "component**s**".

Line 283: "wave energy spectra"
Replace with "wave energy **spectrum**".

Line 289: "2,5 s"
Replace with "2**.**5 s".

Line 294: "already exist and successfully operating and providing"

Replace with "already exist and **are** successfully operating and proving".

Lines 296 and 297: "ships navigating"
Replace with "ship navigation**s**".

Lines 250 to 252: "Further examination and methodology adjustment required together with more data collection during different conditions in the open ocean are required to investigate into these differences."
Rewrite this sentence.

Line 255: "worser"
Replace with "worse"

Line 266: "doesn't"
Replace with "does not".

**References**

Semedo, A., Sušelj, K. and Rutgersson, A., 2009, September. Variability of wind sea and swell waves in the North Atlantic based on ERA-40 re-analysis. In *Proceedings of the 8th European Wave and Tidal Energy Conference, Uppsala, Sweden, 7-10 September* (pp. 119-129).

---

## Referee Comment (RC2)

Title: Wind waves in the North Atlantic from ship navigational radar: SeaVision development and its validation with Spotter wave buoy and WaveWatch III
Author(s): Natalia Tilinina et al.
MS No.: essd-2021-431
MS type: Data description paper

Major Review Comments:

1.  The English language, both grammar and expression, of this manuscript is significantly flawed (some of the example recommendations are provided in the minor comment section below), it should be thoroughly revised.

2.  Use of X-band marine radar for sea surface wind and wave measurements is not new in the literature (see Huang et al., 2017 for examples). More in-depth description of SeaVision, its unique features, and algorithm used to measure the wind waves should be discussed. Comparison with contemporary X-band radar - in design and performance - is recommended.

3.  In lines 180 - 185, and in section 2.3, you mentioned that,
    "$H_s = A + B \sqrt{SNR}$ (1)
    where A and B are empirical calibration coefficients for each radar. In this study calibration coefficients were calculated on the basis of the simultaneous observations with the Spotter wave buoy (see Section 2.3). Calibration coefficients are also used for calculation of the wave energy spectrum. We also use modulation transfer function (MTF, Nieto-Borge et al., 185 2004) to correct radar antenna effects of tilting and shadowing to correct the wave energy spectral density."
    "We further use wave parameters derived from buoy as the "ground truth" for the SeaVision calibration and estimation of the radar calibration coefficients A and B, these coefficients are further used to rescale the SeaVision wave energy spectrum to match buoy spectrum with least squares"

    -   However, nowhere in the paper, the numerical values of 'empirical calibration coefficients' A and B have been given. Please, include those important numbers and describe the calibration procedure more clearly and quantitatively.

4.  From the standard expression of the linear dispersion relation, $\omega^2 = gk \tanh(kh)$, we know that h is the water depth (even the same is given in the reference Nieto-Borge et al., 2004), not directly significant wave height. But in line 165, you claimed that it is the significant wave height which is the key parameter of your results. Therefore, please, review the relevant theory and justify it more clearly.

5.  What quality filters were used? Was there any rain event during any expeditions and data acquisition? Please, discuss these in detail in the data collection section.

6.  The data could not be accessed/retrieved from the given link (https://sail.ocean.ru/tilinina2021/), consequently, the data could not be verified.

7.  X-band radars are usually capable of other wave parameters including sea-swell, which is a very important related parameter. So, authors should justify why the swell measurement was not included in this study.
    7.1.  In the open ocean, swell and surface current contributions to the wave height can be significant depending on the location and time of the year. Therefore, results should be presented on the basis of different sea states. You may use different colors in your scatter plots to indicate different sea states.
    7.2.  How have you estimated significant wave height without swell and surface current information, or how have you separated them?

8.  Validation with the satellite altimeter/SAR or other observational data product (for the possible range) is recommended besides the Spotter wave buoy and WaveWatch model.
    8.1.  Overall description of the WaveWatch III model experiment in section 2.5 is not sufficient. Describe more about the model input, output, and also discuss model limitations. Models usually have their inherent bias/uncertainty, furthermore, the native spatial resolution of ERA5 reanalysis is 31 km. You should include its possible effects on the results.

9.  The focus of this manuscript is validating the SeaVision radar, not the Spotter buoy. So, I recommend presenting "Spotter minus SeaVision (Fig. 5a) and WW3 minus SeaVision (Fig, 5b)" in Figure 5, instead of "Spotter minus SeaVision (Fig. 5a) and Spotter minus WW3 (Fig, 5b)". Same recommendation applies to Figure 7. Also, plot the ground truth along the x-axis, and SeaVision measurement along the y-axis.

10. Solid line must be a 45° line originating from {0,0} in all scatter plots of Figure 6. Quantitative information, i.e., Numerical values of the bias and the STD/root mean square error should be included in the scatter plots (Fig. 6).

11. It is recommended to include the validation results of the wave energy frequency spectrum measured by the SeaVision system in a separate plot.

Minor Review Comments:

1. In line 21, "Simultaneously with SeaVision observations of the wind waves we *were collecting* data in the same locations and time", use simple past tense. Same as in line 99 -- "we were using Spotter wave buoys"; line 120 "we were collecting"; line 121-122 "vessels were drifting".

2. Please, clarify what you mean by 'wind waves' (possibly, in the introduction section) for the general audience and state specifically which wave parameters the SeaVision system measures. Although you mentioned it later in the abstract, "The dataset that supports this paper consists of significant wave height, wave period and wave energy frequency", I think it'd clearer if you mention it at the beginning when you first describe it "In this paper we present the SeaVision system for measuring wind waves' parameters in line 19.

3. The data link can be given in the data section, instead of providing in the abstract.

4. In line 33, "The history of wind waves studies" - should be 'The history of wind wave studies'. Same as in line 114, "the wind waves data", should not be a plural adjective.

5. For lines 33-40, cite proper sources.

6. In line 41, use the simple present for "Remote sensing datasets of the wind waves are *dating back*". Same for lines, 50, 53.

7. In line 41, "when the first satellite radar altimeters missions began measurements of the elevations of the ocean surface" -- should be the first satellite radar altimeter mission.

8. In line 41, "when the first satellite radar altimeters missions began measurements of the elevations of the ocean surface" -- which satellite radar altimeter? Please, cite.

9. In line 44, "Buoys are measuring vertical and horizontal displacements of the ocean surface", -- please use simple present tense instead of progressive.

10. In line 48, "buoys cover only a few locations" -- it is true that buoy networks are sparse for global coverage, nevertheless, it is not "a few".

11. In line 53, "collecting wind waves observations" -- should be 'wind wave observations'.

12. Line 88, "**2 3 Spotter wave buoy data**", please use a dot to indicate a subsection. Same as in line 109 - "**2 1 Expeditions**"; line 125 - "**2 2 SeaVision system**"; "**2 2 1 Radar**

signal preprocessing".; "**2 2 2 Analysis of the sea clutter images**"; "**2 3 Spotter wave buoy data**"; "**2 4 Meteorological data**".

13. Please, use a dot (instead of a comma, which is misleading) to represent fractional numbers, such as in line 117 (59,5°N), line 289 (2,5s), table 1 (231,5) and some other places.

14. Lines 57-58, "(i) collecting wind waves observations in the open ocean using navigational marine X-band radar and (ii) to monitor in real time wave heights, direction and period along the ship track in the open ocean." - use parallel sentences (either gerund or infinitive noth mixed)

15. In line 108, I prefer "2. Data collection and analysis" to "Data collection and analysis principles" as the section heading.

16. In Figure 1, indicate the start, end and direction of the expeditions. For a large portion of the track, especially for figure a and c, data were not collected, why? Please, mention this in the description.

17. In section 2.2.1, and 2.2.2, indentations are used for paragraphs, and nowhere else it is used. Please, make it consistent throughout the paper.
.
18. In line 133, you mentioned "For our purposes we used the shortest possible pulse length of 0.08 μs", please explain why.

19. Please, follow the custom to abbreviate megahertz as MHz in Table 1.

20. Line 157-158, you mentioned, you chose "minimal distance from the ship of 300 m (to avoid potential impact of the ship to the wave field and illumination of the radar signal by the ship).", but for the Spotter wave buoy, in lines 195-196, you mentioned that it was selected to be 200 m. Please, make it consistent. However, if there is any particular reason, please, include your explanation.

21. In line 165, please, correct the unit of gravitational constant 'g' ($ms^{-2}$).

22. In line 195, "200m" vs "300 m" in line 158. Please, make the syntax (space between quantity and unit) consistent throughout the paper.

23. Line 206, "We further use wave parameters derived from buoy" -- please, specify the parameters.

24. Line 228, What is ST6 parameterization? Please, explain ST6 parameterization and the discrete interaction approximation (DIA) scheme a little more about it considering the general audience.

25. Line 255, ''worser" should be worse.

References:

Huang, W., Liu, X., & Gill, E. W. (2017). Ocean wind and wave measurements using X-band marine radar: A comprehensive review. Remote sensing, 9(12), 1261.

Nieto Borge, J., RodrÍguez, G. R., Hessner, K., & González, P. I. (2004). Inversion of Marine Radar Images for Surface Wave Analysis, Journal of Atmospheric and Oceanic Technology, 21(8), 1291-1300. Retrieved Apr 4, 2022

---

## Author Comment (AC1)

Response to Anonymous Reviewer #1

We thank Reviewer #1 for the careful evaluation of the manuscript, constructive comments and the detailed technical suggestions. Below we provide our replies in a point-by-point manner with the responses given in blue and the comments of Reviewer #1 given in italic black.

**R#1 C1:** *Abstract: "Measurements with SeaVision were quality controlled and validated by comparison with Spotter buoy data and WaveWatch III experiments." One of my main criticisms to this work is the use of wave modelling for validation of in-situ observations. It needs to be considered that numerical models provide an approximation of physical phenomena, and they need to be quality controlled against observations, it does not sound logical/scientific to go the other way around.*

AC: We thank Reviewer#1 for this comment. We fully agree with R1 that it is not scientifically correct to say that we use wave modeling results for validation of in-situ observations. We rather argue in the manuscript that WaveWatch III model hindcast is another widely used source of wind wave data which can be compared with SeaVision measurements. This is exactly what has been done in the MS. We adopted changes in the text of the revised manuscript accordingly:

**Changes in manuscript**: Modified text reads as:

….."SeaVision measurements were validated against co-located Spotter wave buoy data and intercompared with the output of WaveWatch III simulations"….

**R#1 C2:** *Section 2.1:I suggest more elaboration of the expeditions in this section considering that the main purpose of this manuscripts is to present the dataset which has been collected during the expeditions. For example, information such as region, location (including approximate coordinates), date (including day of month), name of the vessel, distance sailed, start and end point, stations, instruments used, metocean parameters collected, general depth etc. should be included in the text.*

AC: We thank Reviewer#1 for this comment. Section 2.1 has been completely rewritten. We also added Table 1 which contains general information on research cruises. Information on the locations of the data collection are shown in Appendix A.

**Changes in manuscript**: Section 2.1 now reads as:

Figure 1 demonstrates ship tracks of the three research cruises, during which wind wave data were collected. Research cruises were carried out by IORAS research vessels (R/Vs) *"Academik Sergey Vavilov"* and *"Academik Ioffe"*. Table 1 provides a general information about the cruises and detailed information on the coordinates and dates and is provided in Appendix A. The two cruises in the subpolar North Atlantic (Figure 1a, b) were focused on the regular survey of the 59.5°N oceanographic trans-Atlantic cross-section and cross-sections in the Denmark Strait (Verezemskaya et al., 2021). During these cruises the R/V makes full-depth CTD profiling. The distances between the hydrographic stations vary from ~30 km in the open ocean to a few kilometers near the East Greenland coast with the time allocated for each station (ship is drifting) varying from 2 to 6 hours. Here and later in

the manuscript we determine stations as the locations where wind wave observations were carried out (Table A1). Between the stations the R/V travels at a speed of approximately 6 to 10 kn. During the cruise of R/V *"Academik Ioffe"* in the Kara Sea (Figure 1c), stations were somewhat shorter in time (2-3 hours). During all cruises wave observations were carried out after completing hydrographic profiling. For operating solely SeaVision, the R/V position was strictly stationary being controlled by bow and stern thrusters of the R/V. When SeaVision was used together with the free drifting Spotter buoy, the thrusters were off to provide also free drifting of the R/V. This allowed for measurements of the background wave field by both SeaVision and the Spotter buoy. At each station we first released the Spotter buoy with a supplementary floating buoy dumping cable vibrations. Such design allows for the maintenance of at least 300 m distance between the buoys and the R/V. Then, both buoys were in the free-floating mode for at least 30 min during which the recording was performed by both SeaVision and the Spotter buoy (Figure 4, top panel). Lastly, both buoys were pulled back onboard. The Spotter buoy measured vertical and horizontal displacements starting from its release until being retrieved back onboard. After completing measurements at each station, only the data recorded during the free-floating mode were used for the joint analysis of SeaVision and Spotter buoy records. During all SeaVision and Spotter buoy measurements, standard meteorological parameters were measured using the onboard meteostation.

**Table 1: Research cruises during which the wind waves observations were carried out by R/Vs *"Akademik Sergey Vavilov"* (ASV) and *"Academik Ioffe"* (AI). Adjacent numbers in the first column correspond to the R/V cruise numbers counted from the beginning of the R/V operation.**

| Cruise | Start date and location | End date and location | Distance sailed | Number of stations (with Spotter buoy) |
|---|---|---|---|---|
| ASV50 | 08/08/2020 Kaliningrad Russia | 08/09/2020 Kaliningrad Russia | 10465 km | 21 |
| AI57 | 27/06/2021 Kaliningrad Russia | 02/08/2021 Kaliningrad Russia | 7745 km | 11 |
| AI58 | 08/08/2021 Arkhangelsk Russia | 06/09/2021 Kaliningrad Russia | 10611 km | 16 |

**R#1 C3:** *Figure 1: Explain what the reason is behind measurement gaps as depicted in panels (a) and (c). Briefly explain cruise numbers (50, 57, 58) in the text of the section. For instance, cruise 50 out of how many cruises per month/year aboard RV Academic Survey Vavilov? Indicate what RV stand for. For instance, by adding "research vessel (RV)" in the caption. The published dataset at https://sail.ocean.ru/tilinina2021/ seems to be including the measurements at stations shown with orange dots only (Spotter and SeaVision). I wonder why the rest of the observations (SeaVision only measurements shown in green dots) have not been published? Further, the number of stations shown orange dots are not*

*consistent between this figure, Appendix A, and the published dataset. In total, 52 points are shown Figure 1. However, 50 stations are provided in Appendix A and observations at 48 stations have been published. From Cruise 57, station number 3836 is not included in the dataset but provided in Appendix A. From cruise 50, stations 2771 and 2792 are included in the Appendix but not in the dataset and station 2913 is in the dataset but not in the Appendix.*

**AC**: We thank Reviewer#1 for this comment. The explanation of the gaps between the measurement locations are now provided in section 2.1 together with explanation of numbering of the research cruises. 'R/V' is added in the caption.
We have also explained why only the data sampled in the locations indicated with "orange dots" were published, and also mentioned that interested users can easily request all raw radar data (not co-located with Spotter buoy measurements) can from the authors. We did not publish the remaining observations as the data at these locations were not supported with co-located Spotter buoy observations. Our intention was to provide exclusively data from the stations supported by both data sources. In the future were plan to publish all data collected along with the data from ongoing and underway cruises; this larger set will be already based on the validation with buoy measurements.

We thank Reviewer#1 for noticing inconsistency in the number of stations presented in Figure 1 and in the dataset itself. Sorry for this drawback, now the lists are consistent. We also provide the temporary link for the Reviewers' attention -

https://www.pangaea.de/tok/644c8383ea60396920442184e648ad95714c8d9e with the dataset at PANGAEA, where locations correspond to the list on the Appendix A and orange dots count in Figure 1.

**Changes in manuscript**: Section 2.1 was rewritten accounting for all comments, in particular, caption to the Figure 1 now reads as:

**Figure 1: Ship tracks of the three cruises of the research vessels (R/V) Akademik Sergey Vavilov (a) and Akademik Ioffe (b,c). Green dots indicate locations where only SeaVision radar data were collected, orange dots show the locations for which SeaVision records were co-located with Spotter wave buoy measurements. Cruise numbers are counted from the beginning of the R/V operation.**

**R#1 C4:** *Line 115: "other regular deep ocean observations" What are these "other regular deep ocean observations"? Elaborate more or provide examples.*

**AC**: We thank Reviewe#1 for this comment. Indeed, the style here was awkward making it difficult to get the clear meaning. In the revised MS the sentence was edited.

**Changes in manuscript**: This sentence now reads as:

…"During these cruises the R/V makes full-depth CTD profiling. The distances between the hydrographic stations vary from ~30 km in the open ocean to a few

kilometers near the East Greenland coast with the time allocated for each station (ship is drifting) varying from 2 to 6 hours."….

**R#1 C5:** *Lines 116 to 119: In particular, two cruises in the North Atlantic (Figure 1a,b) are related to regular deep ocean observations at the 59,5°N (Verezemskaya et al., 2021; Falina et al. 2007; Gladyshev et al. 2018, 2019; Sarafanov et al. 2008, 2018) and the Arctic expedition is a part of the IO RAS "Floating University of IORAS" program (Stepanova, 2018). This sentence is confusing. It is providing information of the first two expeditions versus the "program" which the third expedition is associated with. Also, it has already been mentioned at lines 115-116 that all three expeditions are carried out by IO RAS. So, what is the need to repeat it here about the "Arctic expedition"?*

**AC**: Thank you for this comment. The whole section is rewritten.

**Changes in manuscript**: This text fragment now reads as:

..."Research cruises were carried out by IORAS research vessels (R/Vs) "Academik Sergey Vavilov" and "Academik Ioffe". Table 1 provides a general information about the cruises and detailed information on the coordinates and dates and is provided in Appendix A. The two cruises in the subpolar North Atlantic (Figure 1a, b) were focused on the regular survey of the 59.5°N oceanographic trans-Atlantic cross-section and cross-sections in the Denmark Strait (Verezemskaya et al., 2021). During these cruises the R/V makes full-depth CTD profiling. The distances between the hydrographic stations vary from ~30 km in the open ocean to a few kilometers near the East Greenland coast with the time allocated for each station (ship is drifting) varying from 2 to 6 hours. Here and later in the manuscript we determine stations as the locations where wind wave observations were carried out (Table A1). Between the stations the R/V travels at a speed of approximately 6 to 10 kn. During the cruise of R/V "Academik Ioffe" in the Kara Sea (Figure 1c), stations were somewhat shorter in time (2-3 hours). During all cruises wave observations were carried out after completing hydrographic profiling."...

**R#1 C6:** *Line 118: Arctic expedition is a part of the IO RAS". What is "Arctic expedition" referring to? The expeditions have not been named previously.*

**AC**: Thank you for this comment. We removed "Arctic expedition".

**Changes in manuscript**: Now we refer to this expedition as:

…."During the cruise of R/V "Academik Ioffe" in the Kara Sea (Figure 1c), stations were somewhat shorter in time (2-3 hours)."….

**R#1 C7:** *Line 121: "During all stations". It is not clear what the word "stations" is referring to i.e, no information about any station has been previously defined in the text.*

**AC**: We thank Reviewer#1 for this comment. Indeed, we had to better elaborate on the meaning of 'stations' in this context. In the revised version of MS we provide a clear explanation.

**Changes in manuscript**: Sentence added:

..."The distances between the hydrographic stations vary from ~30 km in the open ocean to a few km near the East Greenland coast with time allocated for each station (ship is drifting) varying from 2 to 6 hours. Here and after in the manuscript we determine stations as the locations where wind wave observations were carried out (Table A1)."...

**R#1 C8**: *Lines 122 and 123: "to provide conditions for Spotter buoy wave observations in the free floating mode" Is "free floating" referring to the deployment of spotter buoy? If yes, it needs to be clarified in the text. Also, explain why this deployment option has been targeted.*

**AC**: We thank Reviewer#1 for this comment. In the revised version we clarified the meaning of "free floating" mode and explained the procedure of the buoy deployment in a more clear manner.

**Changes in manuscript**: Following sentences were added:

…."Once the ship is drifting at the location of the measurements, the Spotter buoy was deployed and started drifting away from the ship. Note that the ship drift is always faster compared with that of the buoy, thus, the distance between the buoy and the ship progressively increase. When the distance between the ship and the buoy reached at least 300 m, the "free floating" mode of SeaVision and Spotter buoy operation was initiated for at least 30 minutes as described in section 2.1. The longest free floating mode time period at some stations reached up to 1.5 hours. To ensure homogeneity of the analysis we used 20-minute segments from the "free floating" mode time series for further computations of significant wave height, wave spectra and directional moments"....

**R#1 C9**: *Section 2.2: I think this section can be improved by being organised in a more informative manner. I was expecting a thorough description of SeaVision system and its comparison and superiority against other widely used observational systems such as WaMoS-II as it is indicated in the conclusion that "the commercial oceanographic systems for the wind waves monitoring such as WaMoS II, SeaDarQ and WaveFinder already exist ... [this manuscript's] main aim is to develop in the nearest future a low cost, independently operating and stable system that would allow to broaden observational network for the wind waves".*

**AC**: We thank Reviewer#1 for this comment. In the revised version we provide a detailed explanation of the reasons for which we are developing yet another system for monitoring of wind waves and what are the main conceptual differences between our approach and system design compared to the commercial observational systems such as WaMoS II, SeaDarQ or WaveFinder.

**Changes in manuscript**: The whole Section was rewritten, in particular we added following text:

….."Development of the SeaVision system was based on a commonly accepted approach of the recording and analysis of the sea clutter images. Using a similar approach to commercial systems such as WaMoS II (http://www.oceanwaves.de), SeaDarQ (Greenwood et al., 2018) and WaveFinder (Park et al., 2006) were developed. These commercial systems provide customers with their original software and hardware (sometimes including the X-band radar itself). In our approach we are focused on the development of an independently operating and low-cost system compatible with the existing navigation radars with which ships are already equipped."…..

**R#1 C10**: *Explain why the dataset only includes one dimensional frequency wave spectrum and no information is provided about the directionality of waves. Underway observations and their file types deserve a much more comprehensive explanation considering the scope of ESSD journal.*

**AC**: We thank Reviewer#1 for this comment. Reply to this comment is similar to the one, that we provide for the comment 19 by R#1. Indeed, we provide 1D spectra from both Spotter bouy and SeaVision. There are two reasons why we do not provide data on swell and wind waves separately in this manuscript: (i) the methodology development for accounting contribution of swell and wind waves is still under development in SeaVision and have not been tested to a full extend, and (ii) when a proper algorithm (and likely modification of SeaVision) is developed and the uncertainties of separation of wind waves and swell on the basis of sea clutter images are quantified (expected to be quite large), this will need to be addressed in separate study. We plan to include swell and wind wave separation in the future study and a new versions of SeaVision.

The dataset was reorganized and now we provide information on wave directions together with the other additional parameters.
The temporary link at PANGAEA is - https://www.pangaea.de/tok/644c8383ea60396920442184e648ad95714c8d9e.

The list of parameters available now for the users is the following:

buoy_freq - Spectral frequency of buoy measurements;
buoy_Szz - Vertical Displacement Energy Power Spectra (buoy);
radar_freq - Spectral frequency of radar measurements;
radar_Szz - Vertical Displacement Energy Power Spectra (radar);
Variables below are averaged for the period of Vertical Displacement Energy Power Spectra computation (20 minutes). Radar movement parameters:
radar_lag - Mean Lag Speed of Radar;
radar_gyro - Mean Direction by Gyrocompass 0..360;
radar_sog - Mean Speed Over Ground by GPS;
radar_cog - Mean Course Over Ground by GPS;
radar_sfsds - Spectrum Fitted Ship Drift Speed;

radar_sfsdd - Spectrum Fitted Ship Drift Direction 0..360;
radar_arf - Mean Radar Antenna Rotation Frequency (per minute);
radar_pdir - Wave Spectrum Patch Direction 0..360;
radar_psize - Wave Spectrum Patch Size;
Wave statistics from radar:
radar_hs - Significant Wave Height by Radar;
radar_ts - Mean Wave Period by Radar;
radar_ds - Mean Wave Direction (to) 0..360 by Radar;
Wave statistics from buoy:
buoy_hs - Significant Wave Height by Buoy;
buoy_ts - Mean Wave Period by Buoy;
buoy_ds - Mean Wave Direction (to) 0..360 by Buoy;
Meteorological measurements:
meteo_wspd - Mean Wind Speed;
meteo_wdir - Mean Wind Direction (from) 0..360;
meteo_pres - Mean Atmospheric Pressure;
meteo_temp - Mean Atmospheric Temperature;
meteo_humd - Mean Humidity

**Changes in manuscript**: Data availability section now reads as:

Datasets that contains significant wave height, wave period, wave direction, wave energy frequency spectrum and other related parameters from both SeaVision and the Spotter buoy at the locations of every station (Table A1, Gavrikov et al., 2022) is available through the PANGAEA repository (https://doi.org/10.1594/PANGAEA.939620). Users interested in the analysis of the raw radar dataset or in the wave characteristics in the locations were measurements were carried out only with SeaVision are welcome to request access from Alexander Gavrikov (gavr@sail.msk.ru)

**R#1 C11:** Line 133: For our purposes we used the shortest possible pulse length. "Explain what the reason is for setting the radar to short pulse length i.e., how the resolution of images may be affected etc.

**AC**: We thank Reviewer#1 for this comment. Indeed, the reason for setting the radar to short pulse length was not clearly explained. In the revised version we made the changes to clarify this issue.

**Changes in manuscript**: Explanation of the choice of the short pulse in added to the text of the manuscript:

…."Radars can optionally operate at the pulse lengths of 0.08 μs, 0.25 μs, 0.5 μs, 0.8 μs, 1.0 μs. For our purposes we used the smallest possible pulse length of 0.08 μs (at the so-called "short-pulse" mode - SP1) providing the highest possible resolution of the image (thus the best resolution of the ocean surface). Our X-band radars are characterized by a 3.18 cm wave length of the emitted electromagnetic waves (Table 2). The pulse length is the emission time of the wave beam, thus the number of the emitted waves and the area of the reflection at the ocean surface (defining spatial resolution) increase with increasing pulse length"…..

**R#1 C12:** Line 162: "this results into the three-dimensional spectrum S(kx,ky,f)". Explain what Kx and Ky are.

**AC**: Thank you. Now it is explained.

**Changes in manuscript**: The corresponding text now reads as:

…"This returns for each sector, the three-dimensional spectrum $S_{3d,image}(k_x, k_y, f)$

where $f = \omega/2\pi$ is the frequency (Hz) and $\omega$ is the angular frequency, $k_x$ and $k_y$

(rad/m) are the components of the wave vector $\vec{k}(k_x, k_y)$"….

**R#1 C13, C14 and C15**: *Line 164 and 165: The correct expression of the linear dispersion relation is: =√ tanh ( h). The angular frequency has been previously defined as , why has it been changed to Ω in this equation? The unit of g (gravitational acceleration) should be corrected to m\*s-2 and H should be replaced with h which is the depth, not wave eight!! Also, correct the equation in Figure 3.*

*Line 166: "the signal (ΩBGN)" What is this signal? What does "BGN" stand for?*

*Line 185 and 186: "wave period Ts estimated traditionally using the first moment of the spectrum". Provide the equation for the wave period calculations. I also suggest using the standard annotation of wave period based on the first moment "Tm1" or "Tm01" instead of "Ts".*

**AC**: Thank you for noticing these inconsistencies. In the revised version we corrected the equation for the linear dispersion relation, explained all notations in the equations and also added the equation for the wave period estimation. The whole section 2.2.2 is now rewritten, we also provide new Appendix B, where we explain details of the processing of images in detail.

**Changes in manuscript**: The whole section 2.2.2 was rewritten, Appendix B was added.

**R#1 C16:** *Line 191: "Table A1 provides a list of all locations where SeaVision+Spotter buoy (or SeaVision only) measurements were carried out". From the published dataset, I can see in all the stations provided in Table A1 both SeaVision and Spotter data exist. No data is available at stations with SeaVision only measurements.*

**AC**: We thank Reviewer#1 for this comment. In the published dataset and in Table A1 we only provide data measured simultaneously with SeaVision and Spotter.

**Changes in manuscript**: *(or SeaVision only) removed.*

**R#1 C17:** *Lines 200 and 201: "Analysis of the raw vertical and horizontal displacements recorded by buoy starts from the selecting in timeseries the "free floating" measurements". Explain in the text how free-floating measurements have been identified from timeseries.*

**AC, Changes in manuscript**: Explanation added:

…."The Spotter buoy measured vertical and horizontal displacements starting from its release until getting back onboard. After completing measurements at each station only the data recorded during the free-floating mode were used for the joint analysis of SeaVision and Spotter buoy records"….

**R#1 C18:** *Line 202: "using common definitions (see Appendix in Raghukumar et al., 2019) ". I suggest adding an appendix that provides definition of all the parameters in the manuscript/published dataset.*

**AC, Changes in manuscript**: Thank you. We added Appendix C with all parameters required.

**R#1 C19:** *Line 203: "in the frequency range of the wind waves". I wonder why you have limited yourself to wind waves only, both from the spotter buoy in this the reason behind disregarding swell observations? Couldn't it be included in the dataset/manuscript? If not, it needs to be clarified why. Further, a range of 0.05 to 0.3 Hz seems to be underestimating frequency of wind waves. For example, from Semedo et al., (2009) frequency of wind sea in North Atlantic seems to be above 0.2 to 0.3 Hz in summer when IORAS expeditions have taken place (see Figure 7 at Semedo et al., (2009)). The frequency/ period ranges can also be confirmed from example wave spectrum in Figure 4 of current manuscript. Please also confirm the frequency range applied to the published data (one dimensional frequency spectra) for calculation of integrated wave parameters provided in the "Global Attributes" of the NetCDF files i.e., Hs_radar , Ts_radar, Hs_buoy, Ts_buoy.*

**AC**: We thank Reviewer #1 for this comment. Indeed, in the MS from SeaVision and Spotter buoy we provide only wind wave statistics disregarding swell. At the same time in the dataset we provide 1D spectra from both Spotter bouy and SeaVision. There are two reasons why we do not provide data on swell and wind waves separately in this manuscript: (i) the methodology development for accounting contribution of swell and wind waves is still under development in SeaVision and have not been tested to a full extend, and (ii) when a proper algorithm (and likely modification of SeaVision) is developed and the uncertainties of separation of wind waves and swell on the basis of sea clutter images are quantified (expected to be quite large), this will need to be addressed in separate study. We plan to include swell and wind wave separation in the future study and a new versions of SeaVision.

We also thank Reviewer#1 for noticing inconsistency in the integration frequencies in the equation. Of couse, the range of frequencies in integration is different. We used frequencies from 0.01 to 1.25. This was corrected in the revised text. We confirm that in the published dataset for one dimensional spectra frequencies run from 0.01 - 1.25 for buoy and from 0.09 - 0.41 for SeaVision, this can be now easily seen from parameters buoy_freq and radar_freq in the netcdf file.

**Changes in manuscript**: Sentence on swell separation added to the Data availability section:

…"In this dataset we only provide wind waves statistics, disregarding separation of the swell and wind waves at this stage of the SeaVision development. We plan to include this procedure into the next studies. At the same time, we provide one dimensional spectrum that potentially allows to see first and seconds peaks associated with winds waves and swell (an example is shown in Figure 4)"…

**R#1 C20:** *Figure 4: It should be indicated that this spectrum is an example. Please also provide the station number where this spectrum has been recorded. From the published dataset, I can see that only one-dimensional frequency wave spectra are provided. I am surprised to see the information about wave directions in the bottom right panel. Is there a point I am missing here? It may be better to plot the wave spectrum in frequencies instead of periods.*

*Indicate what PSD stands for. Also, describe what $\theta_p$ and $\sigma\theta_p$ are. I suggest labelling different panels with (a), (b), (c), etc. This also applied to Figures 2 and 3.*

**AC**: We thank Reviewer #1 for this comment. You are right. We only provide 1D spectra and average wave direction without directional spectra estimates. We removed directional spectra estimates from Figure 4 for the consistency between published dataset and plots in the manuscript and provided station numbers and location details.

**Changes in manuscript**: Sentence added:

…"Example of the wave energy spectrum for 20-minute Spotter buoy record is shown in Figure 4"…

Station number provided. Indicated as an example in the Data availability section. Labeling of the vertical axis changed to: power spectral density ($m^2$/ cycles/ second). Labeling added. Figures 2 and 3 are organigrams, so we decided to keep them without labeling.

**R#1 C21:** *Section 2.4: I cannot find any information regarding the meteorological data in the published dataset at* https://sail.ocean.ru/tilinina2021/. *It would be beneficial to publish the meteorological data (after being quality controlled) together with the wave observations.*

**AC**: Thank you for this comment. We included meteorological data in the published dataset.

**Changes in manuscript**: In the Data availability section:

…."Datasets that contains significant wave height, wave period, wave direction, wave energy frequency spectrum, meteorological data and other related parameters"...

**R#1 C22 and C23:** *Section 2.5: A WaveWatch-III model for this study need a much more extensive explanation than one paragraph only. Model physics and packages, setup, calibration, and validation should be comprehensively explained. I am surprised to see that no information is provided about model calibration and validation.*

*Lines 223 to 225: "We run WaveWatch III (WW3DG, version 6.07, WW3) spectral wave model with ERA5 reanalysis (Hersbach et al., 2020) as lateral boundary conditions with 0.1° spatial and 1 hourly temporal resolutions." What does it mean to run the model with ERA5? The model's forcing and boundary conditions (parameters with their temporal and spatial resolution) need to be clearly explained It is not clear whether "0.1° spatial and 1 hourly temporal resolutions" refers to the resolution of forcing and boundary conditions or the actual wave model.*

**AC**: We added more extensive description of the WW3 set up. Thank you for this comment. In the revised version Table 3 with the details of experiments was added.

**R#1 C24:** *Line 238 to 240: "In general, for the lower wind speeds SeaVision underestimates Hs by up to 50 cm and overestimates Hs for the higher wind speeds. This effect can be due to better ripples development on the ocean surface during higher winds affecting the signal to noise ratio (Formula 1)." I am a bit confused here, from Figure 5(a) the difference between Hs observed with Spotter and SeaVision are less, and it increases for wind speeds more that 8 m/s. The text seems to be indicating the opposite while describing better ripple developments at high winds.*

**AC**: We thank Reviewer#1 for noticing this mistake.

**Changes in manuscript**: The whole Section 3 was rewritten.

**R#1 C 25:** *Line 247: "there are two stations (2901 and 2937 see Table A1) where this difference reaches almost 100 cm" Show these points in Figure 5 by a label and/or different colour.*

**AC**: We thank the Reviewer#1 for this suggestion, we highlited in Figure 5 three points, where the differences in significant wave height are larger than 1 m. Additionally, according to the recommendations of the Reviewer#2 we made changes to Figure 5: "Spotter minus SeaVision" and "WW3 minus SeaVision" instead of "Spotter minus SeaVision and "Spotter minus WW3". "Spotter minus SeaVision" (Fig. 5a) differences exceeding 1 m were recorded at stations 2901, 2928, 2937 and for the "WW3 minus SeaVision (Fig, 5b)" at the station 2841.

**Changes in manuscript**: The whole Section 3 is rewritten now. Updated Figure 5 now stands as follows:

[Figure]

**Figure 5: Difference in the significant wave height (H$_s$) estimates for all stations as a function of the wind speed: Spotter buoy ("ground truth") *minus* SeaVision (a), WW3 *minus* SeaVision (b). Dash lines mark the mean difference across all data points. Red squares and circles mark differences higher than 1 m.**

**R#1 C26:** *Figure 6 (and lines 253 to 259): The fitted lines should cross the [0 0] points i.e., logically when Spotter record a wave height (or wave period) equal to zero, SeaVision should return zero, too (it is similarly true about WaveWatch-III estimates). Overestimation of wave height and period by WaveWatch-III against Spotter measurements may be indicating that the model has not been properly calibrated. I suggest including other error statistics such as Root Mean Square Error (RMSE) and Scatter Index (SI). I wonder why the number of points in these plots (~32) is less than the number of stations (~50)? Surprisingly, I get different looking plots from the published dataset while reading the values from "Global Attributes" of each NetCDF file published. Below is the significant wave height from buoy versus radar, for example:*

**AC**: We thank Reviewer#1 for noticing these inconsistencies. We carefully checked and corrected plots to be consistent with the published dataset. Now all points in the panel correspond exactly to the data from the published dataset. We also included RMSE and SI statistics into the plot panels. The fitting line now crosses [0 0] points.

**Changes in manuscript**: Updated Figure 6 looks as:

[Figure]

**Figure 6: Scatterplots of the significant wave height (Hs) and wave period (Tm01) revealed by SeaVision and measured by Spotter (a,c) as well as revealed by SeaVision and simulated with WW3 (b,d) for all stations. Together with Root Mean Square Error (RMSE) and Scatter Index (SI) statistics.**

**Changes in manuscript**:

**R#1 C27:** *Lines 265 and 266: "waves direction (from)" Does it mean the wave directions are in "coming from" convention? It needs to be further explained.*

**AC**: We thank Reviewer#1 for noticing this inconsistency. For the mean wave directions they are coming from the indicated direction. I.e. we use 'meteorological convention'. For the wind it also is coming from (i.e. meteorological definition – 'wind flows into the compass').

**Changes in manuscript**: We added Appendix C with all parameters and clarification of wave and wind directions:

…"We provide definitions of all parameters included in the published dataset in Appendix C. For wind and wave directions we use meteorological convention implying that both wind and waves are coming from the specified direction (blow into compass)"….

**R#1 C28:** *Figure 7: This figure needs a more extensive explanation. The sources of errors and inconsistencies need to be described. Similar to figure 4, I am surprised to see directional wave roses while no information about directionality of waves in provided in the published dataset, please explain.*

**AC**: We added an explanation of the observed differences and also the results of intercomparison with satellite altimeter tracks crossovering the domain were included.

**Changes in manuscript**: This section is rewritten now and additional figures added.

**R#1 C29:** *Data Availability: I suggest including a calendar as well as a metadata in the repository to provide information about the dataset each folder/link includes. Parameters/information available in the dataset and their description (including an extensive description of Global Attributes and Variables in NetCDF files) are worth being added to the manuscript, probably as an Appendix. Also, indicate the full length of time that the measurements are covering.*

**AC**: We thank Reviewer#1 for this comment. In each netcdf file in the repository we provide dates and the other related metadata parameters in the 'Global Attributes' header of netcdf, as shown below:

```
// global attributes:
            :cruise = "AI58" ;
            :ship = "R/V Academik Ioffe" ;
            :station = "3911" ;
            :station_latitude = "70.3680" ;
            :station_longitude = "58.0456" ;
            :station_start_time_UTC = "2021-08-12-13-15-13" ;
            :station_duration_min = "20" ;
            :radar_model = "JRC-JMA-9122-6XA-25kW-6ft" ;
            :radar_ADC = "80MHz" ;
            :antenna_rotation_frequency_perMIN = "24.36" ;
}
```

**Figure R1.** Global attributes for the station 3911 in the netcdf file in the repository. We added Appendix C with variables provided in Repository and their naming.

**Changes in manuscript**: Appendix C added with definitions of parameters.

**R#1:** *Technical corrections*

1. The English language, specifically grammar and punctuations, need to be revised and corrected throughout the text

The text of the manuscript was significantly reworked and entirely proofread. English grammar and punctuations were corrected.

2. Consistent symbols and annotations should be used in the text and figures. For example, the WaveWatch-III wave model is indicated by others. Other examples are: Line 175: "WaveWatch III" in some parts and "WW3" in Significant wave height being indicated by "H" and "Hs", and Figure being indicated by "Fig." and "Figure".

All these incosistensies were corrected.

We thank Reviewer#1 for his careful look onto conceptual and technical issues and evaluation of the maniscript. As the text of the manuscript was reworked significantly we adopted all technical suggestions everywhere, except for text pieces which were entirely rewritten. Below we provide reponces to the minor comments, omitting comments adressed to rewritten pieces of the text.

**Line 135**: "SeaVision system (Fig. 2) connected to the radar via splitter, **it** digitizes and records directionally". Replace with "SeaVision system (Fig. 2)**,** connected to the radar via splitter, digitizes and records directionally"
**Response**: Thank you. Corrected.

**Line 176**: "spectra power". Replace with "spectral power"
**Response**: Thank you. Corrected.

**Line 190**: "waves observations" Replace with "wave observations"
**Response**: Thank you. Corrected.

**Line 199**: "we use". Replace with "we used"
**Response**: Thank you. Corrected.

**Line 279:** Nowadays there is still exists gap. Correct the sentence grammatically
**Response**: Thank you. Corrected. In the manuscript now this sentence reads as:

…."Ocean wind waves play a critically important role in air-sea energy and gas exchanges (Gulev and Hasse 1998; Andreas et al. 2011; Blomquist et al. 2017; Ribas-Ribas et al. 2018; Cronin et al. 2019; Xu et al. 2021 among many others) and in ocean surface mixing (McWilliams and Fox-Kemper 2013; Buckingham et al. 2019; Studholme et al. 2021), thus being an important active component of the coupled climate system (Cavaleri et al. 2012; Fan and Griffies 2014). At the same time, massive long-term observations of wind waves over global oceans still have insufficient coverage and quality compared to other surface variables (e.g. air and sea surface temperatures)."….

**Line 279:** Wind waves is. Replace "with wind waves **are**"

**Response**: Thank you. Corrected.

**Line 280:** "component". Replace with "components"
**Response**: Thank you. Corrected.

**Line 283:** "wave energy spectra". Replace with "wave energy spectrum"
**Response**: Thank you. Corrected.

**Line 289:** "2,5 s". Replace with "2.5 s"
**Response**: Thank you. Corrected.

**Lines 250 to 252**: "Further examination and methodology adjustment required together with more data collection during different conditions in the open ocean are required to investigate into these differences." Rewrite this sentence.
**Response**: Thank you. The sentence is completely removed.

---

## Author Comment (AC2)

Response to Dr. Alamgir Hossan (Reviewer #2)

We thank Dr. Alamgir Hossan for careful evaluation of the manuscript and valuable suggestions. Below we provide our replies in a point-by-point manner with our responses given in blue and comments of Dr. Alamgir Hossan in italic black.

**AH C1:** *The English language, both grammar and expression, of this manuscript is significantly flawed (some of the example recommendations are provided in the minor comment section below), it should be thoroughly revised.*

**AC**: According to this comment and also comments by Reviewer#1 the revised MS was entirely edited and proofread with many being fully re-written. We are very much hopeful that English of the revised version meets the ESSD standards.

**AH C2:** *Use of X-band marine radar for sea surface wind and wave measurements is not new in the literature (see Huang et al., 2017 for examples). More in-depth description of SeaVision, its unique features, and algorithm used to measure the wind waves should be discussed. Comparison with contemporary X-band radar - in design and performance – is recommended.*

**AC**: Thank you for this comment. In the revised version we provided more extensive description of the SeaVision advantaged and methodology with specifications of the technical details of the hardware and methodological details of algorithms.

**Changes in manuscript**: The whole Section focused on methodology for the analysis of sea clutter images was rewritten, we have also added Appendix B which gives detailed explanations of associated algorithms. Specifically, the following text elaborating the purposes of SeaVision development was added:

…..”Development of the SeaVision system was based on a commonly accepted approach of the recording and analysis of the sea clutter images. Using a similar approach to commercial systems such as WaMoS II (http://www.oceanwaves.de), SeaDarQ (Greenwood et al., 2018) and WaveFinder (Park et al., 2006) were developed. These commercial systems provide customers with their original software and hardware (sometimes including the X-band radar itself). In our approach we are focused on the development of an independently operating and low cost system compatible with the existing navigation radars with which ships are already equipped”…..

**AH C3:** *In lines 180 - 185, and in section 2.3, you mentioned that,m"Hs = $A + B\ SNR$ (1) where A and B are empirical calibration coefficients for each radar. In this study calibration coefficients were calculated on the basis of the simultaneous observations with the Spotter wave buoy (see Section 2.3). Calibration coefficients are also used for calculation of the wave energy spectrum. We also use modulation transfer function (MTF, Nieto-Borge et al., 185 2004) to correct radar antenna effects of tilting and shadowing to correct the wave energy spectral density." "We further use wave parameters derived from buoy as the "ground truth" for the SeaVision calibration and estimation of the radar calibration coefficients A and B, these coefficients are further used to rescale the SeaVision wave energy spectrum to match buoy spectrum with least squares" - However, nowhere in the paper, the*

*numerical values of 'empirical calibration coefficients' A and B have been given. Please, include those important numbers and describe the calibration procedure more clearly and quantitatively.*

**AC**: We thank Dr. Alamgir Hossan for this comment. In the revised version we provide a clear explanation of the calibration procedures in section 2.2.2 which was re-written. We also provide important details of the computations in Appendix B.

**Changes in manuscript**: Values of the calibration coefficients A and B added in the Table 2:

| Calibration coefficients A and B | A = -0.4042, B = 1.0034 | A = -0.4042, B = 1.0034 |
| --- | --- | --- |

**AH C4:** *From the standard expression of the linear dispersion relation, $\omega 2 = gk$ $\tanh(kh)$, we know that h is the water depth (even the same is given in the reference Nieto-Borge et al., 2004), not directly significant wave height. But in line 165, you claimed that it is the significant wave height which is the key parameter of your results. Therefore, please, review the relevant theory and justify it more clearly.*

**AC**: Thanks for noticing this inconsistency. In the revised version we corrected the equation for the linear dispersion relation, explained all notations in the equations and also added the equation used for wave period estimation. As we pointed out above, the whole section 2.2.2 is now rewritten, we also provide new Appendix B, where we explain the details of the processing of images in detail.

**Changes in manuscript**: The whole section 2.2.2 was rewritten, Appendix B was added.

**AH C5:** *What quality filters were used? Was there any rain event during any expeditions and data acquisition? Please, discuss these in detail in the data collection section.*

**AC**: We thank Dr. Alamgir Hossan for this comment. We did not use any quality filters as the standard output of the meteorological station was already quality controlled, the Spotter wave buoy data do not require any quality control, as the buoy system passed calibration procedure. SeaVision and radar operation indeed can be affected by the rain events as the raindrops can scatter electomagneitc radar signal. We checked our records with respect to this and found no rain events during observation periods.

**Changes in manuscript**: Sentence on weather conditions was added in the lines 240-241 of the revised version of the manuscript:

….."We note that local weather conditions, specifically rain events, can potentially affect the electromagnetic radar signal as the raindrops absorb and scatter radar signal. However the analysis of current weather has shown that no rain events were observed during observations."….

**AH C6:** *The data could not be accessed/retrieved from the given link (https://sail.ocean.ru/tilinina2021/), consequently, the data could not be verified.*

**AC**: Thank you for noticing this. The temporary link at PANGAEA repository is available now for the Reviewers' attention -

https://www.pangaea.de/tok/644c8383ea60396920442184e648ad95714c8d9e

**AH C7:** *X-band radars are usually capable of other wave parameters including sea-swell, which is a very important related parameter. So, authors should justify why the swell measurement was not included in this study. In the open ocean, swell and surface current contributions to the wave height can be significant depending on the location and time of the year. Therefore, results should be presented on the basis of different sea states. You may use different colors in your scatter plots to indicate different sea states. How have you estimated significant wave height without swell and surface current information, or how have you separated them?*

**AC**: We thank Dr. Hossan for this comment. Indeed, we provide only wind wave statistics disregarding swell. At the same time, in the dataset, we provide 1D spectra from both Spotter buoy and SeaVision system. There are two reasons why we do not provide data on swell and wind waves separately in this manuscript: (i) the methodology development for accounting contributions from swell and wind waves is still under development in SeaVision and have not been tested to a full extent, and (ii) when a proper algorithm (and likely modification of SeaVision) is developed and the uncertainties of separation of wind waves and swell on the basis of sea clutter images are quantified (expected to be quite large), this will need to be addressed in separate study. We plan to include swell and wind wave separation in the future study and a new versions of SeaVision. At this stage we do calculate surface currents, however we are aware that methodologies for surface currents are successfully used in e.g. WaMoS system.

**Changes in manuscript**: Comment on swell separation was added to the Data availability section:

…"In this dataset we only provide wind waves statistics, disregarding separation of the swell and wind waves at this stage of the SeaVision development. We plan to include this procedure into the next studies. At the same time we provide one dimensional spectrum that allows to see first and second peaks associated with winds waves and swell"…

**AH C8:** *Validation with the satellite altimeter/SAR or other observational data product (for the possible range) is recommended besides the Spotter wave buoy and WaveWatch model.*

**AC**: We thank Dr. Alamgir Hossan for this comment. We added the results of intercomparison of our wind wave observations with all available satellite altimeter crossovers.

**Changes in manuscript**: Figures 8 and 9 added to the text of the MS. Section 3 significantly reworked, intercomparison with satellite altimeters crossovers added.

**AH C8:** *Overall description of the WaveWatch III model experiment in section 2.5 is not sufficient. Describe more about the model input, output, and also discuss model limitations. Models usually have their inherent bias/uncertainty, furthermore, the native spatial resolution of ERA5 reanalysis is 31 km. You should include its possible effects on the results.*

**AC**: We thank Dr. Alamgir Hossan for this comment. We added more extensive description of the WaveWatch III set up. Also in the revised version Table 3 with the details of experiments was added.

**AH C9:** *The focus of this manuscript is validating the SeaVision radar, not the Spotter buoy. So, I recommend presenting "Spotter minus SeaVision (Fig. 5a) and WW3 minus SeaVision (Fig, 5b)" in Figure 5, instead of "Spotter minus SeaVision (Fig. 5a) and Spotter minus WW3 (Fig, 5b)". Same recommendation applies to Figure 7. Also, plot the ground truth along the x-axis, and SeaVision measurement along the y-axis.*

**AC**: Thank you for this suggestion. We swapped Spotter minus WW3 to WW3 minus SeaVision in Figure 5 and the results should be more clear now. In addition, we highlighted extreme differences reaching 1 m with red on both plots.

**Changes in manuscript**: Updated Figure 5 now stands as follows:

[Figure]

**Figure 5: Difference in the significant wave height (H$_s$) estimates for all stations as a function of the wind speed: Spotter buoy ("ground truth")** *minus* **SeaVision (a), WW3** *minus* **SeaVision (b). Dash lines mark the mean difference across all data points. Red squares and circles mark differences higher than 1 m.**

**AH C10:** *Solid line must be a 45° line originating from {0,0} in all scatter plots of Figure 6. Quantitative information, i.e., Numerical values of the bias and the STD/root mean square error should be included in the scatter plots (Fig. 6).*

**AC**: Thank you for this comment. We included RMSE and SI statistics into the plot panels (also suggested by Reviewer#1). The fitting line now crosses [0 0] points.

**Changes in manuscript**: Updated Figure 6 looks as:

[Figure]

**Figure 6: Scatterplots of the significant wave height (H$_s$) and wave period (T$_{m01}$) revealed by SeaVision and measured by Spotter (a,c) as well as revealed by SeaVision and simulated with WW3 (b,d) for all stations. Together with Root Mean Square Error (RMSE) and Scatter Index (SI) statistics.**

**AH C11:** *It is recommended to include the validation results of the wave energy frequency spectrum measured by the SeaVision system in a separate plot.*

**AC**: Thank you for this comment. In the dataset, that supports this manuscript we provide the digital data quantifying 1D wave energy spectrum which can be easily plotted from the netcdf files (Figure R1). Example of the buoy spectra is also provided in Figure 4.

[Figure]

**Figure R1.** 1D wave energy spectra on the basis of Spotter buoy and SeaVision.

**Changes in manuscript**: We highlighted in the text of the MS that 1D spectra are available by adding the sentence in the Data availability section:

…"At the same time we provide one dimensional spectrum that potentially allows to see first and seconds peaks associated with winds waves and swell"…

We thank Dr. Alamgir Hossan for his efforts focused on the careful evaluation of our manuscript. As the text of the manuscript was significantly reworked and many text fragments were rewritten, we adopted all minor comments everywhere, except for text pieces which were entirely rewritten.

*Minor comments*

**AH**: *"Simultaneously with SeaVision observations of the wind waves we were collecting data in the same locations and time", use simple past tense. Same as in line 99 -- "we were using Spotter wave buoys"; line 120 "we were collecting"; line 121-122 "vessels were drifting".*
**Response**: Thank you. Corrected.

**AH:** *Please, clarify what you mean by 'wind waves' (possibly, in the introduction section) for the general audience and state specifically which wave parameters the SeaVision system measures. Although you mentioned it later in the abstract, "The dataset that supports this paper consists of significant wave height, wave period and wave energy frequency", I think it'd clearer if you mention it at the beginning when you first describe it "In this paper we present the SeaVision system for measuring wind waves' parameters in line 19.*

**Response**: Thank you for this comment. In the Introduction we provide a clear definition of wind waves along with references highlighting wind waves importance for different applications:…."Wind waves are wind-driven ocean surface gravity waves". We also added to the manuscript Appendix C with all definitions of the dataset parameters, thus making it easier to go through the manuscript. We mention in the beginning of the Section 2 that all definitions of all parameters in the published dataset can be found in the Appendix C.

**AH:** *The data link can be given in the data section, instead of providing in the abstract.*
**Response**: Thank you for this comment. It is a requirement of the ESSD journal to provide link to the dataset that supplies the manuscript both in the abstract and in the Data section.

**AH:** *In line 33: "The history of wind waves studies" - should be 'The history of wind wave studies'. Same as in line 114, "the wind waves data", should not be a plural adjective.*
**Response**: Thank you. Corrected.

**AH:** *For lines 33-40, cite proper sources.*
**Response**: Thank you. Citations are provided.

**AH:** *In line 41: use the simple present for "Remote sensing datasets of the wind waves are dating back". Same for lines, 50, 53.*
**Response**: Thank you. Corrected.

**AH:** *In line 41, "when the first satellite radar altimeters missions began measurements of the elevations of the ocean surface" -- should be the first satellite radar altimeter mission.*
**Response**: Thank you. Corrected.

**AH:** *In line 41, "when the first satellite radar altimeters missions began measurements of the elevations of the ocean surface" -- which satellite radar altimeter? Please, cite.*

**Response**: Thank you. Now the citations are provided as follows: …"when the first satellite radar altimeter missions (Seasat in 1978 (the first satellite to provide data) and Geosat in 1985)"...

**AH:** *In line 44:* "Buoys are measuring vertical and horizontal displacements of the ocean surface", -- please use simple present tense instead of progressive.
**Response**: Thank you. Corrected.

**AH:** *In line 48: "buoys cover only a few locations" -- it is true that buoy networks are sparse for global coverage, nevertheless, it is not "a few".*
**Response**: Thank you. Now this is rewritten as follows: …."However, buoy networks are sparse with most deployments being in the coastal regions and can only effectively serve for verification of all other dataset rather than for developing global or regional climatologies."….

**AH***: In line 53: "collecting wind waves observations" -- should be 'wind wave observations'.*
**Response**: Thank you. Corrected.

**AH:** *Line 88:* "2 3 Spotter wave buoy data", please use a dot to indicate a subsection. Same as in line 109 - "2 1 Expeditions"; line 125 - "2 2 SeaVision system"; "2 2 1 Radar signal preprocessing".; "2 2 2 Analysis of the sea clutter images"; "2 3 Spotter wave buoy data"; "2 4 Meteorological data"
Response: Thank you. This was corrected throughout of the whole manuscript.

**AH:** *Please, use a dot (instead of a comma, which is misleading) to represent fractional numbers, such as in line 117 (59,5°N), line 289 (2,5s), table 1 (231,5) and some other places.*
Response: Thank you. This was corrected throughout of the whole manuscript.

**AH:** *Lines 57-58, "(i) collecting wind waves observations in the open ocean using navigational marine X-band radar and (ii) to monitor in real time wave heights, direction and period along the ship track in the open ocean." - use parallel sentences (either gerund or infinitive noth mixed)*
Response: Thank you. The sentence is rewritten as reads now as: …"We present the design and pre-processing methodology of the SeaVision system along with the dataset collected during the three research cruises (Fig. 1)"….

**AH:** In line 108, I prefer "2. Data collection and analysis" to "Data collection and analysis principles" as the section heading.
Response: Thank you. Corrected.

**AH:** *In Figure 1, indicate the start, end and direction of the expeditions. For a large portion of the track, especially for figure a and c, data were not collected, why? Please, mention this in the description.*
Response: Thank you for this comment. We added Table 1 with the description of the research cruises. We have also significantly reworked the description of the strategy of the field experiments. The locations of the measurements were chosen on the basis of predefined hydrographic stations. This is now clearly posed in the text in lines 125-133.

**AH:** *In section 2.2.1, and 2.2.2, indentations are used for paragraphs, and nowhere else it is used. Please, make it consistent throughout the paper.*
Response: Thank you. Corrected throughout the whole manuscript.

**AH:** *In line 133, you mentioned "For our purposes we used the shortest possible pulse length of 0.08 μs", please explain why.*
Response: Thank you for this comment. Indeed, the reason for setting the radar to short pulse length was not clearly explained in the previous version. In the revised version we made the changes to clarify this issue with the following: …."Radars can optionally operate at the pulse lengths of 0.08 μs, 0.25 μs, 0.5 μs, 0.8 μs, 1.0 μs. For our purposes we used the smallest possible pulse length of 0.08 μs (at the so-called "short-pulse" mode - SP1) providing the highest possible resolution of the image (thus the best resolution of the ocean surface). Our X-band radars are characterized by a 3.18 cm wave length of the emitted electromagnetic waves (Table 2). The pulse length is the emission time of the wave beam, thus the number of the emitted waves

and the area of the reflection at the ocean surface (defining spatial resolution) increase with increasing pulse length".....

**AH:** Please, follow the custom to abbreviate megahertz as MHz in Table 1.
**Response**: Thank you. Done.

**AH:** *Line 157-158, you mentioned, you chose "minimal distance from the ship of 300 m (to avoid potential impact of the ship to the wave field and illumination of the radar signal by the ship).", but for the Spotter wave buoy, in lines 195-196, you mentioned that it was selected to be 200 m. Please, make it consistent. However, if there is any particular reason, please, include your explanation.*
**Response**: Thank you for noticing this inconsistency. Numbers are corrected now.

**AH:** *In line 165, please, correct the unit of gravitational constant 'g' (ms-2).*
**Response**: Thank you. Corrected.

**AH:** *In line 195: "200m" vs "300 m" in line 158. Please, make the syntax (space between quantity and unit) consistent throughout the paper.*
**Response**: Thank you. Corrected through the whole manuscript.

**AH:** *Line 206, "We further use wave parameters derived from buoy" -- please, specify the parameters.*
**Response**: Thank you. This is specified now in the Data availability section as: ..."Datasets that contains significant wave height, wave period, wave direction, wave energy frequency spectrum, meteorological data and other related parameters from both SeaVision and the Spotter buoy at the locations of every station"....

**AH:** *Line 228, What is ST6 parameterization? Please, explain ST6 parameterization and the discrete interaction approximation (DIA) scheme a little more about it considering the general audience.*

**Response**: Thank you for this comment. In the revised version of the manuscript we provide a reference to the WaveWatch III development group basic publication and few other references giving the description of source term (ST6) package for parameterizations of wind input, wave breaking, and swell dissipation and of Discrete Interaction Approximation (DIA) parametrizations.

**AH:** *Line 255, "worser" should be worse.*
**Response**: Thank you. Corrected.

---

## Author Comment (AC3)

Response to Prof. Ian Young

We thank Prof. Ian Young for evaluation of the manuscript and useful comments and suggestions. Below we provide our replies in a point-by-point manner with the responses given in blue and the comments of Prof. Ian Young given in italic black.

**IY 1:** *The English language expression needs to be improved. There were some sections where it was difficult to understand exactly what had been done because of the English expression.*

AC: We thank Prof. Ian Young for this comment. The manuscript underwent proofreading and many text pieces were entirely rewritten. We are very much hopeful that now English grammar and punctuations meet standards of ESSD.

**IY 2:** *Line 92: "SeaVision can be used for operational monitoring of the current wind waves' field for individual ships and continuous collection…" – I don't understand what this means? Do you mean SeaVision can be used to measure currents and wind waves? Which it can.*

AC: Thank you for this comment. We edited this sentence in the revised version. Now it reads as follows:

…."We present the design and pre-processing methodology of the SeaVision system along with the dataset collected during three research cruises (Fig. 1)"….
We also should mention that in the present configuration we have not tested an ability of the SeaVision for measurement of the currents. This is planned for the future separate studies.

**IY 3:** *Line 164: "ocean waves (Fig. 3): Ω = sqrt($gk$ tanh($kH$)), where k is the wave number absolute value (rad/m), g is the gravity acceleration (m•s-1) and H is significant wave height. – this statement is incorrect. In the linear dispersion relation, H is the water depth. I hope this is just a typo and it has not really been applied as written.*

AC: Thank you for the comment to this mistake. Of course, it has not been applied in practice and did not affect the computations themselves. In the revised version we corrected this typo and now the equation is given as below:

$$\omega = \sqrt{gk} + kU cos\theta \ ,$$

where $k$ is the wave number (rad/m), $g$ is gravity (m/s$^2$), $U$ is the surface velocity (m/s) which includes surface current velocity and ship drift, and $\theta$ is the angle between the wave vector $\vec{k}$ and velocity vector $\vec{U}$.

**IY 4:** *On the figure significant wave height (I assume this is what it is) is written as H0, above in the text it is Hs and as noted in Point 3, erroneously as H. Can you please use just one symbol for significant wave height. I suggest Hs.*

AC: Thank you for noticing this inconsistency. In the revised version we corrected the usage of symbols throughout of the whole manuscript.

**IY 5:** *On the bottom left panel there is a 1D spectrum with the horizontal axis as period. On the bottom right panel there is a directional spectrum with the radial distance as frequency. This makes comparison very difficult. Please express all spectra in terms of frequency, as is normally done in the literature.*

AC: Thank you for this comment. We swapped the horizontal axis as periods in terms of frequencies in Figure 4, we also removed directional spectra estimates for a better consistency across figures and dataset output parameters in the MS (directional spectra were not provided).

**IY 6:** *Line 225: What wind was used to force the WW3 model? ERA5?*

AC: WW3 model was forced by 1-hourly wind speeds and sea ice concentrations from ERA5 reanalysis with spatial resolution of ~27 km (Hershbach et al., 2020). It is now stated in the revised version of the manuscript.

**IY 7:** *Conclusions: I was expecting some attempt to explain the observed differences between the buoys and the radar. Is it the difference between a measurement at a point compared to a measurement over a region etc? I think the reader needs some suggestions as to the reasons for the observed differences.*

AC: Thank you for this comment. We considerably reworked Section 3 and provided reasonable explanations of the observed differences. In particular the following text was added in the revised version of manuscript:

…"Overall, the analysis of significant wave heights among these three sources of data (Spotter, SeaVision and WW3) shows that the highest $H_s$ values are measured by the Spotter buoy, lowest are simulated by WW3, with SeaVision being in between. These results are intuitively correct as wave buoys measure the actual elevations of ocean surface, SeaVision provides a proxy of local wave conditions from image analysis (thus imposing averaging over the domain) and is not expected to be as accurate as wave buoy data."….

---

## Author Comment (AC4)

Response to Dr. Vladimir Karaev

We thank Dr. Vladimir Karaev for evaluation of the manuscript and useful suggestions. Below we provide our replies in a point-by-point manner with the responses given in blue and the comments of Vladimir Karaev given in italic black.

*VK: About formula (1). These coefficients (A and B) are unique properties of each radar and constant or it is necessary to do a calculation of coefficients for every experiment? Which physical background for such approximation? Please, give more information.*

AC: Thank you for this comment. In the revised MS we provide the description of the methodology for calculation of the calibration coefficients A and B along with the physical background behind these computations and the actual values of these coefficients given in Table 2. All these details are now given in Section 2.2.2 of the revised MS.

*VK: Fig. 5 and Fig. 6. It is interesting to see a result of comparison for each cruise separately. Is there a difference?*

AC: We thank the Dr. Vladimir Karaev for this suggestion, we added information on the comparisons of the significant wave heights differences across different cruises separately for each cruise in Table 4:

**Table 4: Differences in significant wave height estimates for the three cruises.**

| Mean difference in $H_s$ (m) | ASV50 | AI57 | AI58 |
|---|---|---|---|
| Spotter - SeaVision | 0.27 | 0.05 | -0.06 |
| WW3 - SeaVision | -0.24 | -0.24 | -0.36 |

*VK: Fig. 4. It is no enough for comparison. Is it possible to compare the wave spectrums (radar, buoy and WWIII)?*

AC: Thank you for this comment. In the dataset, which supports this manuscript we provide digital data for 1D wave energy spectrum which can be easily plotted from the netcdf files. In this manuscript our goal was to present SeaVision system together with the dataset of wind wave observations rather than analyse the nature of the differences between these three data sources. Thus, while we do not provide extensive analysis of the spectrum estimates on the basis of the different data sources, we nevertheless discuss in details differences in wind wave parameters for each location pointing to e.g. large drift cases.